# **Experimental Study of Time-averaged Flow and Turbulence Structures over Low-Angle Tidal Dunes under Steady Bidirectional Flows**

Kevin Bobiles<sup>1,2</sup>, Bernhard Kondziella<sup>3</sup>, Christina Carstensen<sup>3,4</sup>, Elda Miramontes<sup>1,2</sup>, Ingrid Holzwarth<sup>3</sup> and Alice Lefebyre<sup>1</sup>

<sup>1</sup>MARUM – Center for Marine Environmental Sciences, University of Bremen, Bremen, Germany

10 Correspondence to: Kevin Bobiles (kbobiles@marum.de)

Abstract. Large-scale, high-resolution flume experiments were conducted under representative steady bidirectional flows in a large recirculating flume to investigate the time-averaged flow and turbulence structures over two-dimensional (2D) fixed tidal dunes. Two dune morphologies were considered for representing asymmetric dunes with low to intermediate-angle slopes. The dune morphologies are an idealised representation of natural tidal dunes. Specifically, the study aims to characterise in detail the influence of dune morphology on the properties of flow separation zones and turbulence structures above these dunes subjected to bidirectional flows, which are a representation of tidal flows. Results show that a smaller permanent flow separation zone is found over the tested intermediate-angle dune (mean lee slope of 12°, steep face of 22 compared to those found over angle-of-repose dunes detected close to the bed where a larger intermittent flow separation is found. The corresponding turbulent wake expands downwards to the trough before dissipating further downstream. Over the tested low-angle dune (mean lee slope of 10°, steep face of 15°), both small permanent and intermittent flow separations are observed. When the flow is opposed to the dune asymmetry and flows over a very gentle side (4°) with a short steep portion (10°), only a very small intermittent flow separation is detected. Over only a straight gentle side (4°), both flow separations are nonexistent. Neither intermediate- or low-angle dunes exhibit a distinct turbulent wake when the flow is opposed to the dune asymmetry, and a turbulence structure similar to that under a flatbed condition can be observed. Large-scale turbulence associated with both types of dunes is observed through the high occurrence of energy containing ejection and sweep events as revealed by quadrant analysis. Overall, our study demonstrates the significant impact of dune morphology, particularly the mean lee slope and steep face on the emerging flow and turbulence structures above intermediate- to low-angle tidal dunes.

## 1 Introduction

The coastal environment is a highly dynamic and active portion of the Earth's surface where the ocean meets the land. Under the action of river, tidal, wind and wave-driven currents, sediments can be mobilised and frequently forms both small- and

<sup>&</sup>lt;sup>2</sup>Faculty of Geosciences, University of Bremen, Bremen, Germany

<sup>&</sup>lt;sup>3</sup>Federal Waterways Engineering and Research Institute (BAW), Hamburg, Germany

<sup>&</sup>lt;sup>4</sup>Present address: Federal Waterways and Shipping Administration (WSV), Kiel, Germany

large-scale rhythmic undulations, collectively known as bedforms. Dunes are large (decimeters to meters in height, meters to hundreds of meters in length) sandy bedforms which are particularly abundant in lower reaches of rivers (Lange et al., 2008), estuaries (Aliotta and Perillo, 1987; Bradley et al., 2013) and in coastal tidal environments (Damen et al., 2018), where they develop into large fields with complex morphologies. Bedforms are considered main drivers of flow resistance and sediment transport in most sandy rivers and coastal estuarine environments (Best, 2005; Coleman and Nikora, 2011; Venditti, 2013) and, thus, a detailed characterisation of flow and turbulence over bedforms is necessary as a first step towards the understanding of many fundamental and practical processes such as hydro-morphodynamic interactions (Villard and Kostaschuk, 1998; Kostaschuk et al., 2004; Herrling et al., 2021), waterways management (Nasner et al., 2009), subsea cable burial and offshore constructions.

40

35

Depending on their morphology, dunes can be classified as high-, intermediate- or low-angle dunes (Lefebvre and Cisneros, 2023) (Fig. 1). High-angle dunes have a lee slope steeper than around 25°, their height is usually on the order of 1/6 of the water depth (Bradley and Venditti, 2017) and they are commonly found in small rivers and laboratory flumes (Venditti et al., 2005; Naqshband et al., 2014) where a unidirectional current is the dominant flow condition. Intermediate dunes have lee slopes between ca. 10 and 25° and are often found in large rivers. On the other hand, low-angle dunes with lee slopes of less than 10° are mostly found in large tidal rivers, estuaries and tide-dominated shelves where a bidirectional tidal current is the dominant flow condition (Nasner, 1974; Aliotta and Perillo, 1987; Lefebvre et al., 2021).

The flow and turbulence dynamics above dunes differ depending on the type of dune (Fig. 1). The overall characteristics of the mean flow and turbulence structures over high-angle dunes are already well documented (Nelson et al., 1993; Mclean et al., 1994; Bennett and Best, 1995; Venditti and Bennett, 2000; Best, 2005). Above high-angle dunes, flow accelerates over the stoss side until it reaches the crest and decelerates over the lee side forming a permanent flow separation zone where a reverse flow is observed. A shear layer, which separates the flow recirculation cell from the overlying undisturbed flow, forms and expands. Along this shear layer, instabilities in the form of Kelvin-Helmholtz instabilities develop giving rise to a large turbulent wake. This turbulent wake expands upward and advects over the stoss side of the adjacent dune. Below this turbulent wake, a newly formed internal boundary layer develops, with a logarithmic velocity profile. Because of the resulting flow structure, a maximum horizontal velocity located at the crest develops and is expected to generate high bottom shear stress capable of generating bedload and suspended sediment transports contributing to the morphodynamic changes of bottom topography.

60

55

Some studies on low to intermediate-angle dunes have been documented in the past (Kostaschuk and Villard, 1996; Roden, 1998; Best and Kostaschuk, 2002; Sukhodolov et al., 2006) although the flow and turbulence structures above these dunes are still not fully investigated. Over intermediate-angle dune, it is surmised that there is rarely a permanent flow separation and it is likely that an intermittent flow separation forms (Lefebvre and Cisneros, 2023). Consequently, the

resulting turbulent wake will differ from that observed for high-angle dunes. Over low-angle dunes, studies (Kostaschuk and Villard, 1996; Roden, 1998; Carling et al., 2000; Best and Kostaschuk, 2002; Sukhodolov et al., 2006; Kwoll et al., 2016) have pointed out the lack of permanent flow separation and only an intermittent flow separation may be expected to form over the lee side (Lefebvre and Cisneros, 2023). The resulting turbulence structure is expected to be weak and might be similar to that over a flatbed condition.

Clearly, the morphology of the dune has a direct influence on the resulting flow and turbulence dynamics above these dunes. The steep face is the lee side slope downstream of the crest that is steeper than the mean lee slope and its adjacent segments. Specifically, the slope and location of the steep face are critical in the formation and properties (i.e., shape, extent, magnitude) of different flow separation zones (i.e., permanent, intermittent or nonexistent flow separation) (Fig. 1). Further evidence highlights that the steep face properties in addition to the mean lee slope dictates the presence and size of flow separation and turbulent wake (Lefebvre et al., 2016; Lefebvre and Cisneros, 2023).

Figure 1: Types of dunes based on their morphology.

100

River and tidal dunes have contrasting shapes (Fig. 2). In large rivers, low-angle river dunes have rounded crest with their steepest lee slope close to the trough (Lefebvre et al., 2016; Cisneros et al., 2020). Low-angle tidal dunes, on the other hand, have sharp pointed crest with their steepest slope close to the crest (Dalrymple and Rhodes, 1995; Lefebvre et al., 2021). Such low-angle tidal dunes possessing a sharper crest and steep slope close to the crest have not been explored extensively and, therefore, how different the flow properties over such dunes are compared to high-angle dunes or low-angle river dunes is currently not well known.

In addition, the complex interaction between reversing tidal flows and the natural morphology of tidal dunes has not been considered to a great extent. Recently, it has been demonstrated that there is a need for a more extensive study to further elucidate the flow dynamics above these low-angle tidal dunes (Carstensen and Holzwarth, 2023).

Figure 2: Schematic representations of river and tidal dunes.

In this study, we conduct large-scale flume experiments to provide detailed descriptions of the flow and turbulence properties over representative two-dimensional tidal dunes under idealised steady bidirectional flows. The modelled dunes are intermediate to low-angle dunes (

length of 220 m consisting of two straight sections connected by a semi-circular segment at both ends (Fig. 3). The straight channel section has a length of 70 m, a width of 1.5 m and a maximum water depth of 1.3 m. The flow is generated by a bow thruster pump located in an underground pipeline at the opposite side of the straight channel section. A maximum flow velocity of 1 m/s can be imposed in either directions by reversing the pump flow direction. Carstensen and Holzwarth (2023) showed that secondary flows at both ends of the straight channel section can be attributed to geometric configurations such as bends or curves in the channel and to fluctuating turbulence associated with the pump generation itself. In order to avoid the influence of secondary flows, measurements were taken far away (around 30 m) from the bends (Fig. 3).

Figure 3: Schematic diagram of the recirculating flume.

# 2.2 Modelled tidal dunes and hydrodynamic conditions

The shapes and dimensions of the laboratory dunes are depicted in Fig. 5 and their morphological properties are summarised in Table. 1. The dunes used in this study were modelled following the dunes found in the tidal river part of the Weser, upstream from the estuarine part, in a region without salt water but influenced by tidal currents (Lefebvre et al., 2021). The dimensions and slopes of the field tidal dunes are summarised in Table 1. Intermediate angle field dunes with maximum steep side angle of 22.5° and mean steep side angle of 12° account for 10% of the observed dunes in the Weser. These dunes are expected to generate considerable turbulence and, thus, have a strong influence on flow resistance. Low angle field dunes with maximum steep side angle of 14.2° and mean steep side angle of 8.5° are the dominant type of dune (48%) observed in the Weser. These dunes are only expected to generate intermittent flow separation and turbulence with intensities proportional to the slope of the steep face. The laboratory dunes are obtained by applying a Froude scaling with a length scale of 1:15. Specifically, laboratory dune 1 is an idealised representation of an intermediate-angle tidal dune and laboratory dune 2 is an idealised representation of a low-angle tidal dune. Both laboratory dunes maintained a constant bedform height to water depth ratio (H/h) of 0.15 and bedform aspect ratio (H/L) of 0.05 consistent with previous observations (Bennett and

Best, 1995; Bradley and Venditti, 2017). These nondimensional parameters ensure that the generated velocity field is representative of the flow above fully developed bedforms.

The relevant morphological parameters of a tidal dune are depicted in Fig. 4. F1 and F2 represent ebb and flood flows, respectively. H is the bedform height defined as the vertical distance from the trough to the crest and L is the bedform length defined as the horizontal distance between either crest or troughs of adjacent dunes. The steep side is the side of the dune where the flow is aligned with the dune asymmetry (i.e., F1 flow). The gentle side, on the other hand, is the side of the dune where the flow opposes the dune asymmetry (i.e., F2 flow). We refer to "flow aligned with the dune asymmetry" when the flow is going over the steeper side of the dune and "flow opposed the dune asymmetry" when the flow is going over the gentler side of the dune. The corresponding mean and maximum angles are also shown in Fig. 4. Steep face height, H<sub>SF</sub>, is defined as the vertical projection of the steep face.

Figure 4. Schematic diagram of tidal dune morphology.

The hydrodynamic conditions such as water levels, flow velocities and flow directions in the Weser Estuary change during every ebb-flood and spring-neap tidal cycle and, thus, flow velocity and water levels vary following various cycles (Lange et al., 2008; Lefebvre et al., 2021). Since our flume experiments are not intended to reproduce the full tidal conditions in the Weser, this study simply adopts representative flow conditions that are comparable to the field conditions. Steady bidirectional flows, defined as F1 and F2 depth-averaged flow velocities, are imposed for both directions with a magnitude of 0.3 m/s. The mean water depth is set to 1.0 m. These scaled down conditions are based on the maximum flow velocity during ebb phase at the Weser which is about 1.0 m/s and the mean water depth is about 14 m at the area of Weser where intermediate to low-angle dunes are observed (Lefebvre et al., 2021; Carstensen and Holzwarth, 2023). The flow conditions at the laboratory are scaled down through Froude scaling based from the observed hydrodynamic conditions at the Weser.

The resulting Froude number (Fr =  $U/\sqrt{(gh)}$ ) is the same for both scales implying dynamic similarity between field and laboratory dunes. The hydrodynamic conditions at both field and laboratory scales are also summarised in Table 1.

# 2.3 Experimental setup

The two laboratory dunes and the setup inside the recirculating flume are shown in Fig. 5. Ten fixed concrete dunes with a fine-sandblasted surface to provide a natural grain roughness were placed inside the flume. Measurements were conducted in two opposite flow directions (i.e., F1 and F2 flows) for the two laboratory dunes. For each measurement, the mean water depth, dune length, dune height and flow velocity were kept constant to allow comparison between measurements. In order to measure flow properties in equilibrium with the dune morphology, a field of 10 identical dunes were deployed at the centreline of the straight channel section of the flume, covering a total distance of 30 m, and measurements were taken for each flow direction above the 5<sup>th</sup> dune (Fig. 5).

Figure 5. Laboratory tidal dunes and experimental flume setup.

Table 1. Dune morphology and hydrodynamic conditions between field and laboratory dunes.

| Parameters            | Intermediate-<br>angle field<br>dunes | Low-angle<br>field dunes | Laboratory<br>dune 1:<br>intermediate-<br>angle | Laboratory<br>dune 2: low-<br>angle |
|-----------------------|---------------------------------------|--------------------------|-------------------------------------------------|-------------------------------------|
| Bedform height (H), m | 2.3                                   | 1.5                      | 0.15                                            |                                     |
| Bedform length (L), m | 50.9                                  | 45.5                     | 3.0                                             |                                     |

| Bedform aspect ratio (H/L)               | 0.05             | 0.04 | 0.05              |    |
|------------------------------------------|------------------|------|-------------------|----|
| Mean steep side angle (°)                | 11.9             | 8.5  | 12                | 10 |
| Max. steep side angle (°)                | 22.5             | 14.2 | 22                | 15 |
| Mean gentle side angle (°)               | 4.5              | 3.6  | 4                 | 4  |
| Max. gentle side angle (°)               | 10               | 9    | 4                 | 10 |
| Mean water depth (h), m                  | 14               | 13.8 | 1                 |    |
| Height-to-depth ratio (H/h)              | 0.16             | 0.11 | 0.15              |    |
| Depth-ave. velocity (F1 = F2 = U), $m/s$ | 1                |      | 0.3               |    |
| Froude number                            | 0.08             |      | 0.1               |    |
| Reynolds number                          | $1.4 \cdot 10^7$ |      | 3·10 <sup>5</sup> |    |

# 2.4 High-resolution flow measurements

Velocity measurements were conducted using a sideward looking Nortek Acoustic Doppler Velocimeter (ADV), Vectrino (Nortek, 2018).

For each flow and dune set up, around 1300-1700 velocity measurements (around 70-90 velocity profiles), each for 2 minutes with a sampling rate of 100 Hz, were conducted (Fig 6). To facilitate quick data acquisition and accurate positioning of the ADV, a motorised metal framed structure (motion unit) was installed on top of the flume, which automatically moved the instrument to a pre-determined location. From here on, we adopt the following conventions for consistency and quick reference to each experimental test. The coordinate axes follow the right-hand rule where positive horizontal x-axis is pointing to the right, positive horizontal y-axis is pointing inward towards the page and positive vertical z-axis is pointing upward. The velocity components are defined as u, v, and w for x, y and z axes, respectively. 'DUNE1' and 'DUNE2' refer to laboratory dune 1 and laboratory dune 2, respectively (Fig. 5). 'F1' refers to flow condition when the flow is aligned with the dune asymmetry and 'F2' refers to flow condition when the flow is opposed to the dune asymmetry. For instance, 'DUNE1\_F1' is the test when the flow is aligned with the dune asymmetry above laboratory dune 1.

Figure 6. ADV point measurement layout. a) DUNE1\_F1, b) DUNE1\_F2, c) DUNE2\_F1, d) DUNE2\_F2. F1: flow aligned with dune asymmetry; F2: flow opposed to dune asymmetry.

## 2.5 Data processing and analysis

Velocity measurements were processed to generate a high-resolution two-dimensional velocity field and other derived flow and turbulence properties. After despiking the data (Goring and Nikora, 2002) and performing further quality control, the instantaneous velocities (u, v and w) were separated into their mean and fluctuating components based on Reynolds decomposition. That is, for instantaneous horizontal component u,

$$u = \overline{u} + u' \tag{1}$$

where  $\overline{u}$  is the mean (time-averaged) component of the velocity and u' is the fluctuating component of the velocity. The same decomposition can also be done to the v and w velocity components. The vertical gradient of the mean horizontal velocity,  $\partial \overline{u}/\partial z$ , is calculated from the mean component of the horizontal velocity,  $\overline{u}$ .

The intermittency factor, IF, which is an indicator of how frequent flow reversal occurs, is calculated from the instantaneous horizontal velocity, u as

$$IF = \frac{N_{reversed\_u}}{N_{total}} \cdot 100 \tag{2}$$

where  $N_{reversed\_u}$  is the count in the velocity measurements when the instantaneous horizontal velocity, u, changes its direction from positive to negative flow (i.e., u 

where  $u'_{rms}$  and  $w'_{rms}$  are the root-mean square values of the horizontal and vertical components of the velocity, respectively. Corresponding with previous studies (Bennett and Best, 1995; Best and Kostaschuk, 2002; Kwoll et al., 2016), a hole size (HS) of 2 is used to better delineate significant quadrant events from background turbulence, which is especially relevant for quadrants 2 and 4 since they are often classified as positive contributors to Reynolds stress (Bennett and Best, 1995) and sediment transport (Unsworth et al., 2018).

## 3 Results

## 3.1 Characteristics of time-averaged flow over intermediate- and low-angle dunes

The time-averaged flow fields for both dunes with F1 and F2 flow setups are shown in Fig. 7. Typical zones of deceleration and acceleration can be observed above the steep side and gentle side, respectively, under the F1 flow setup (Figs. 7a and 7c). This pattern reverses under the F2 flow setup (Figs. 7b and 7d). Flow acceleration is pronounced just before and at the crest for both dunes and for both flow setups. A rapid flow deceleration occurs at the steep face of both dunes under both flow setups. These observations confirm the influence of topographic forcing on the time-averaged flow field.

Figure 7. Time-averaged flow fields. a) DUNE1\_F1, b) DUNE1\_F2, c) DUNE2\_F1, d) DUNE2\_F2.

A permanent flow separation zone can be observed very close to the dune surface for both dunes over the steep face, when the flow is going over the steeper dune side (Fig. 8). Both flow separations start to develop at or shortly after the brink point,

which marks the beginning of the steep face, and are extending over the steep dune side. Although both dunes show the presence of flow separation, their sizes are different. Over DUNE1, the flow separation length is  $L_{FSZ} = 2.3 H$  or  $3.0 H_{SF}$ . The maximum thickness is  $Th_{FSZ} = 0.14 H$  or  $0.18 H_{SF}$ . Over DUNE2, the flow separation length and thickness are much shorter and thinner with dimensions of  $L_{FSZ} = 1.25 H$  or  $3.74 H_{SF}$  and  $Th_{FSZ} = 0.06 H$  or  $0.17 H_{SF}$ . Note that both dunes have the same bedform height but different steep face heights.

Figure. 8. Permanent flow separation zones. a) DUNE1\_F1, b) DUNE2\_F1.

In order to determine the extent of permanent and intermittent flow separation, the positions of the lines showing the 0% and 50% intermittency factor are calculated (Fig. 9). For DUNE1\_F1, an intermittent flow separation extending the entire steep side of the dune is detected. Below this intermittent flow separation, a permanent flow separation is limited to the steep face of the dune. Above DUNE2\_F1, the intermittent flow separation becomes almost limited to the steep face and the permanent flow separation is significantly limited in extent compared to DUNE1\_F1.

When the flow is opposed to the dune asymmetry and over the straight gentle side (4°), no permanent flow separation is detected (Fig. 9). When the gentle slope is made of only one segment (DUNE1\_F2), there is also no intermittent flow separation. However, when a small steeper slope of 10° is present (DUNE2\_F2), a small intermittent flow separation is observed.

Figure 9. Intermittency factor, IF (%). a) DUNE1\_F1, b) DUNE2\_F1, c) DUNE1\_F2, d) DUNE2\_F2. Note different scale used for Fig. 9d.

Over DUNE1\_F1, a large and wide horizontal velocity gradient,  $\partial \overline{u}/\partial z$ , is observed to develop past the brink point and dissipate downstream of the steep side and over the gentle side of the next dune (Fig. 10a). The thickest portion of the velocity gradient can be found at the steep face. This steep velocity gradient indicates the presence of a shear layer and significant vorticity. Above DUNE2\_F1, a thin shear layer, almost attached to the bed and with a diffused-like structure, is observed (Fig. 10b). For DUNE1\_F2, a very thin and weak shear layer can be seen to develop very close to the bed (Fig. 10c). Above DUNE2\_F2, characteristics of an attached shear layer similar to that of DUNE1\_F2 but with a slightly steeper velocity gradient within the small portion of the steeper slope can be detected (Fig. 10d).

Figure 10. Vertical gradient of time-averaged horizontal velocity,  $\partial \overline{u}/\partial z$  (/s). a) DUNE1\_F1, b) DUNE2\_F1, c) DUNE1\_F2, d) DUNE2\_F2.

## 3.2 Characteristics of turbulence structures over intermediate- and low-angle dunes

For the case of DUNE1\_F1, a well-defined turbulent wake can be seen developing just downstream of the brink point above the steep face (Fig. 11a). It propagates downstream of the steep side until it dissipates over the gentle upstream side of the next dune. The strongest portion of the wake is observed immediately downstream of the steep face. Over DUNE2\_F1, the turbulence is further reduced with no defined wake structure and a more diffuse pattern can be detected (Fig. 11b). High TKE is concentrated within the immediate vicinity of the bed and diminishes towards the upper portion of the water column.

Although not as strong as the previous case (Fig. 11a), high TKE occurs within the trough and the immediate downstream portion of the gentle side. When the flow is opposing the dune asymmetry and is flowing over the gentle side, a similar trend of TKE is observed for both dunes (Figs. 11c & d). High TKE is concentrated in the near-bottom region of the flow with no appreciable wake structure. The TKE in the near-bottom flow region diminishes further when there is no steep face present at all (Fig. 11c).

Figure 11. Turbulent kinetic energy, TKE (m²/s²). a) DUNE1\_F1, b) DUNE2\_F1, c) DUNE1\_F2, d) DUNE2\_F2. Note different scale used for Fig. 11a.

The spatially-averaged Reynolds stresses generally increase towards the bed with rapid increase starting from the crest level and downwards from the zero mean bed elevation (Fig. 12). The spatially-averaged Reynolds stresses are high when the flow is aligned with the dune asymmetry (i.e., F1 flow setup, Figs. 12a & c). When the flow is opposed to the dune asymmetry (i.e., F2 flow setup), the spatially-averaged Reynolds stress profiles for both dunes have an almost comparable vertical structure (Figs. 12b & d). The spatially-averaged Reynolds stress profile is maximum for the case of DUNE1\_F1 (Fig. 12a).

out in previous studies (Nelson et al., 1993; Bennett and Best, 1995; Nikora et al., 2001; Mclean et al., 2008; Kwoll et al., 2016) by performing a regression analysis through the linear segment of the profile and projecting it downwards towards the zero mean bed elevation. The bed shear stress estimation shows a high coefficient of determination for all the linear fits (Fig. 12). The linear fit is done above elevations of 0.2 m, 0.34 m, 0.3 m and 0.35 m for DUNE1\_F1, DUNE1\_F2, DUNE2\_F1 and DUNE2\_F2, respectively (Fig. 12). The corresponding total bottom shear stresses, τ<sub>o</sub>, are 0.38 Pa, 0.04 Pa, 0.06 Pa and 0.04 Pa, showing a reduction of total bottom shear stress for lower mean slopes. The local peak of the spatially-averaged Reynolds stress profile is located around the zero mean bed elevation for all cases although its magnitude is much larger for DUNE1\_F1 compared with the other cases indicating the significant influence of dune morphology on the local distribution

The spatially-averaged Reynolds stress profile can also provide a direct estimate of the total bottom shear stress as pointed

of total bottom shear stress.

Figure 12. Spatially-averaged Reynolds stress profiles, <τ<sub>uw</sub>> (Pa). Note different scale for Fig. 12a.

# 3.3 Characteristics of turbulent events (Quandrant Analysis)

The results of quadrant analysis (Fig. 13) depict the percentages of observations for the four quadrant events. Percentages of observations for outward interaction (Q1) and wallward interaction (Q3) events show an increasing trend above the bed. Elevated percentages for these two events occur within z = 0.4-0.8 m above the bed. On the other hand, percentages of observations for ejection (Q2) and sweep (Q4) events are highest near the bed and mid water column, especially for ejection events, and decrease toward the surface. Among the four quadrant events, the highest percentage of observations are observed mostly for ejection (Q2) events. High sweep (Q4) occurrences mainly take place in the very near-bottom flow region.

A very intense and high occurrence of ejection (Q2) events are detected at the steep face of the intermediate-angle dune and are being brought up further into the water column toward the water surface (DUNE1\_F1). These high Q2 occurrences ejected from the steep face are merging to a broader region of high Q2 occurrence located within z = 0.3-0.7 m from the bed. This pattern seems to occur also for the low-angle dune (DUNE2\_F1) although it is not as pronounced as the previous dune. The spatial distribution of high ejection (Q2) occurrences seems to change when the flow is opposed to the dune asymmetry (F2 flow setup). High Q2 occurrences are observed concentrating at the mid-water column around z = 0.2-0.4 m above the bed. Furthermore, high sweep (Q4) occurrences are also observed diminishing at the very near bottom when the flow direction changes.

Figure 13. Percentage of observation at each significant quadrant event (%), HS = 2.0. a) Outward interaction event, Q1, b) Ejection event, Q2, c) Wallward interaction event, Q3, d) Sweep event, Q4.

# 3.4 Schematic representation of flow and turbulence structures above intermediate- and low-angle dunes

The conceptual diagram presented here demonstrates the significance of dune morphology and flow bidirectionality on the emerging flow and turbulence properties above tidal dunes. (Fig. 14). Specifically, the presence and slope of the steep face are the factors controlling whether a permanent, intermittent or nonexistent flow separation will form. A permanent flow separation exists for intermediate-angle dunes and even for low-angle dunes as long as a steep face (>15°) is present. The properties and extent of the resulting flow separation are specific to the properties of the steep face (i.e., steep face angle). Above this permanent flow separation is an intermittent flow separation that covers a wider region and whose properties and extent are also proportional to the properties of the steep face.

Over the intermediate-angle dune with the flow aligned with the dune asymmetry, a small and elongated permanent flow separation develops above the steep face. This permanent flow separation is mostly populated by sweep (Q4) events which are dominant in the near-bottom flow region. The intermittent flow separation develops over the entire steeper side of the dune up until the lower portion of the gentler side of the next dune. A defined turbulent wake is generated along the flow separation due to the instabilities and expansion of the developing shear layer. This nearly attached turbulent wake expands downwards and advects further downstream into the next dune. Furthermore, this wake region is populated by energetic ejection (Q2) events which rise further upward in the water column and merge with the wake from upstream dunes. This stacking of wakes demonstrates the influence of bedform fields (i.e., upstream dunes) in the resulting turbulence dynamics in the downstream dunes.

Over the low-angle dune, a similar structure to that of intermediate-angle dunes can be observed except that the shape of the intermittent flow separation is narrow and elongated. There is also no defined turbulent wake that can be detected but only a region of high TKE populated by ejection events. The inner flow region which contains considerable turbulence shifted downward within the water column indicating that these turbulence structures have become more limited within the lower portion of the water column.

Flow bidirectionality (i.e., flow is opposed to the dune asymmetry) also alters the flow and turbulence dynamics above these dunes. Permanent flow separation is nonexistent over gentle lee sides (4°). Interestingly, an intermittent flow separation exists even if the mean slope of the dune is very gentle provided a steep portion (10°) is present. Regions of steep velocity gradients and high TKE are concentrated very close to the bed leading to dampening of the shear layer and turbulent wake. Consequently, large-scale turbulence structures are attenuated due to less vigorous and less frequent ejection (Q2) and sweep (Q4) events compared to dunes with a steeper slope.

Finally, our conceptual diagram emphasizes that turbulence production and, thus, turbulence structures are not solely determined by flow separation. This is especially apparent when flow bidirectionality is considered. A sufficient velocity gradient is already enough for macroturbulence generation.

Figure 14. Conceptual diagram of flow and turbulence dynamics over intermediate- and low-angle tidal dunes.

#### 4 Discussion

#### 4.1 Flow separation zones

Previous studies have pointed out the absence of permanent flow separation over low-angle dunes (Smith and Mclean, 1977; Kostaschuk and Villard, 1996; Roden, 1998; Carling et al., 2000; Best and Kostaschuk, 2002) and the possible presence of intermittent flow separation (Carling et al., 2000; Best and Kostaschuk, 2002). The present findings demonstrate that both permanent flow separation and intermittent flow separation can exist for intermediate and low-angle dunes depending on the lee side morphology, in particular the presence of a steep slope.

While permanent flow separation is well documented over steep asymmetric high-angle dunes (Nelson et al., 1993; Bennett and Best, 1995; Roden, 1998; Kwoll et al., 2016), the permanent flow separation detected in this study show contrasting characteristics with typical large permanent flow separation (Best, 2005; Venditti, 2013; Lefebvre et al., 2014a, 2016). The 380 observed permanent flow separation is more elongated and limited in extent. This small permanent flow separation only occupies the near-bottom flow region very close to the bed. Similar to previous observations above high-angle dunes (Bennett and Best, 1995; Kostaschuk, 2000), a small region at the steep face characterised by upward vertical velocity can also be detected. Because of the limited extent of the flow separation above intermediate- and low-angle dunes, the flow separation lengths are much shorter compared to previously reported values typically between 4-6H for high-angle dunes (Engel, 1981; 385 Paarlberg et al., 2007; Lefebvre et al., 2014; Naqshband et al., 2014), 4.3H-6.5H<sub>SF</sub> for estuarine dunes (Carstensen and Holzwarth, 2023), 2.1-4.1H for 2D river dunes (Kwoll, 2013; Kwoll et al., 2016) and 5H<sub>SF</sub> for 3D river dunes (Lefebvre, 2019). The difference in the lengths of permanent flow separation can be attributed to the properties of the steep face (i.e., location and slope angle) as pointed out in previous studies (Lefebvre et al., 2016; Lefebvre, 2019; Lefebvre and Cisneros, 2023). The presence of a steep face is a controlling factor on the generation of flow separation. Over a steep slope, a stronger adverse pressure gradient (i.e.,  $\partial p/\partial x >> 0$ ) is encountered by the mean flow leading to a stronger and larger flow expansion which cause a permanent boundary layer separation. Such a process is also pointed out in a previous study about high and low-angle river dunes (Kwoll et al., 2016). On the contrary, only a weaker adverse pressure gradient is encountered over the gentle side of the dune which is not enough to form a permanent flow separation.

The intermittent flow separations that have been observed in this study have not been covered in much detail in previous studies. Specifically, we are able to show the presence and extent of intermittent flow separation for intermediate- and low-angle dunes. This study also confirms the previous claim that over low-angle dunes, an intermittent flow separation is present (Carling et al., 2000; Best and Kostaschuk, 2002) regardless of whether a permanent flow separation exists. Furthermore, our results demonstrate that even for low-angle dune possessing a very gentle mean slope but with some steeper portion, an 400 intermittent flow separation can form.

420

425

This study has also provided insights into the influence of flow bidirectionality on the flow and turbulence dynamics above dunes, particularly over intermediate- and low-angle dunes which are rarely considered in past studies. This has implications for natural dunes since actual flow conditions vary over time due to variables such as tidal flows and river discharge.

## 4.2 Turbulent wakes

There are no universally accepted criteria in defining the turbulent wake. For instance, Lefebvre and Cisneros (2023) defined the turbulent wake as the zone where the TKE is twice the mean TKE observed over a flatbed configuration with same flow conditions (i.e. flow velocity and water depth). Studies have also defined the turbulent wake based on a threshold value such as the TKE that is at least 70% of the maximum TKE (Lefebvre et al., 2014a; Lefebvre et al., 2014b; Carstensen and Holzwarth, 2023). Other studies have also defined the turbulent wake as the region characterised by high frequency of ejection (Q2) and sweep (Q4) events (Unsworth et al., 2018). Some have also related the formation of a turbulent wake with the shear layer, pointing out that the turbulent wake is the result of the advected free shear layer which finally diffuses downstream carrying high turbulence intensities, Reynolds stresses and creating a wake structure that is similar to that of wake past a cylinder (Mclean et al., 1994; Bennett and Best, 1995). While most of the above-mentioned literature has used the TKE distribution to define the turbulent wake, the streamwise turbulence intensity (I<sub>u</sub>) and Reynolds stress have been also used to define the wake. Specifically, the isolines of I<sub>u</sub> = 1.25 and TKE = 2 Pa are said to be a good indicator of defining the turbulent wake (Venditti, 2007).

In this study, we adopt the definition that the turbulent wake is the region enclosed by the isoline of 70% of the maximum TKE and once the wake has been delineated, the extent of the wake is defined as the horizontal distance between the farthest ends of this wake. Since there can be an isolated isoline that passes the threshold, we further restrict our determination of the wake length to consider only the largest contiguous isoline in the TKE distribution. Based on this definition and with a maximum TKE of  $0.0089 \text{ m}^2/\text{s}^2$  for DUNE1\_F1, the turbulent wake extent,  $L_{\text{wake}}$ , is estimated to be 2.4H or  $3.13H_{\text{SF}}$  (Fig. 15). This turbulent wake is smaller compared to, for instance,  $5.5H_{\text{SF}}$  (Carstensen and Holzwarth, 2023), 10-17H (Maddux et al., 2003) and  $13 H_{\text{SF}}$  (Lefebvre, 2019). This suggests that tidal dunes tend to have shorter wake lengths than river dunes owing to their lower mean slopes and gentler steep face.

Figure 15. Turbulent wake (70%TKE<sub>max</sub> isolines) over DUNE1 F1.

In contrast to previous studies where the turbulent wake is located high above the dune surface (Best, 2005; Venditti, 2013; 430 Lefebvre et al., 2014a; Lefebvre et al., 2014b; Carstensen and Holzwarth, 2023), the turbulent wake detected in this study is very close to the bed and looks almost attached to it. This is because the shear layer is also very close to the bottom. This finding confirms the intricate relation between the shear layer and the turbulent wake. The observed turbulent wake is also weaker in magnitude compared to that over high-angle dunes (Lefebvre et al., 2014a; Kwoll et al., 2016). Although there is no appreciable turbulent wake for the other cases in this study on the basis of the 70% TKE<sub>max</sub> isoline, the overall distribution of the TKE seems to follow a near-wall wake structure similar to that over flatbed conditions (Kline et al., 1967).

# 4.3 Presence and generation of large-scale turbulence structure above intermediate- and low-angle dunes

The quadrant analysis results demonstrate that most of the observed large-scale turbulence structures result from the highly energetic and frequent ejection (Q2) and sweep (Q4) events. Specifically, ejection (Q2) events are dominant in the shear layer and turbulent wake regions. Sweep (Q4) events are mostly confined in the near-bottom flow especially in the trough section and in the newly formed internal boundary layer. These findings demonstrate the capability of intermediate- and low-angle dunes to generate large-scale turbulence structures (Kostaschuk and Church, 1993; Best and Kostaschuk, 2002) although their intensity and frequency are weak compared to that over high-angle dunes (Nelson et al., 1993; Bennett and Best, 1995). These findings also imply that the presence of large-scale turbulence might be attributed to intermittent flow separation and does not entirely depend on the presence of permanent flow separation (Best and Kostaschuk, 2002).

Our results suggest that these large-scale turbulences are generated through the shedding of the Kelvin-Helmholtz instabilities along the shear layer (Bennett and Best, 1995; Kadota and Nezu, 1999). This is especially pronounced from the

spatial distribution of ejection (Q2) events for both dunes although this is weaker and less intense for the case of low-angle dunes. The absence of permanent flow separation while there is still a presence of large-scale turbulence suggests that these structures can be generated if there is a sufficient velocity gradient capable of developing a strong shear layer (Best and Kostaschuk, 2002).

Moreover, this large-scale turbulence is characterised by ejection (Q2) and sweep (Q4) events which are more significant for the case of the intermediate-angle dunes than for the low-angle dunes. The bidirectional flows alter the structure of the large-scale turbulence for both the intermediate- and low-angle dune as evident from the patterns of occurrences of the ejection and sweep events.

This observed large-scale turbulence might also suggest some impact on energy exchange and sediment transport even for the case of intermediate- and low-angle dunes. Ejection (Q2) and sweep (Q4) events feed energy into these turbulence structures by extracting energy from the mean flow via positive contributions to the Reynolds stress (Bennett and Best, 1995; Best and Kostaschuk, 2002; Unsworth et al., 2018). A positive Reynolds stress together with the velocity gradient are the key components for turbulence production. This can also imply that the energy exchange between mean flow and turbulence above low-angle dunes is still enough to generate large-scale turbulence structures. The positive contribution of ejection and sweep events on turbulence has an influence on sediment transport as pointed out in previous studies (Kostaschuk and Church, 1993; Unsworth et al., 2018). For instance, some studies have found out that the upwelling motion of slower fluid particles from the near-bottom flow region caused by the highly frequent significant ejection (Q2) events present in the shear layer and turbulent wake regions are responsible for the observed elevated suspended sediment concentration on both the crest and lee side of the dune (Thorne et al., 1989; Kostaschuk and Church, 1993). On the other hand, the presence of frequent in-rush velocity directed towards the bottom is said to be more responsible for the mobilisation of the more coarser sedimentary materials (i.e., bedload transport) (Thorne et al., 1989).

# **5** Concluding remarks

High-resolution, large-scale flow measurements were conducted over representative intermediate- to low-angle tidal dunes to provide detailed descriptions of the time-averaged flow properties and turbulence structures under bidirectional steady flows. The following conclusions can be drawn from this study:

1. Permanent and intermittent flow separation zones exist for the two tested dune configurations. The properties of the flow separation are directly influenced by the presence and slope of the steep face and mean lee side angle. Specifically, we are able to quantify in detail the shape and extent of both permanent and intermittent flow separations for both intermediate- and low-angle dunes.

- 2. Over the intermediate-angle dune, the elongated permanent flow separation is short and almost attached to the bed. These characteristics are in contrast with the large and wide permanent flow separation observed for high-angle dunes. An intermittent flow separation that scales with the dune height and covers a wide extent is observed above the small permanent flow separation.
- 3. Over the low-angle dune, both permanent and intermittent flow separations are detected although their shape and extent are considerably reduced in comparison to the intermediate-angle dune configuration.
- 4. A distinct turbulent wake is generated above the intermediate-angle dune. The turbulent wake is almost attached to the bed and expands downward before dissipating further over the downstream dune. The wake length is found to be shorter than that of high-angle dunes.
  - 5. No defined wake structure is observed over the tested low-angle dune. Instead, high TKE is concentrated in the very near-bottom region of the flow with a structure similar to a diffuse pattern observed over flatbed conditions.
- 6. Quadrant analysis results demonstrate the occurrence of large-scale turbulence even for intermediate- and low-angle dunes through the presence of strong and frequent ejection and sweep events.
- 7. Flow bidirectionality alters the flow and turbulence dynamics for both tested dunes. There is no permanent flow separation for both dunes when the flow is opposed to the dune asymmetry. Interestingly, even if the flow is going over a very gentle slope (4°), an intermittent flow separation can still form provided a small steep (10°) segment is present. Also, a defined turbulent wake does not form for both dunes when the flow direction changes.
- 8. Finally, our study highlights the capability of intermediate- and low-angle dunes to generate permanent flow separation in contrast to previous claims that it is nonexistent for these types of dunes. Moreover, the results also show that large-scale turbulence is present even in the absence of a permanent flow separation. This implies that a sufficient velocity gradient capable of developing an energetic shear layer is indeed enough to generate large-scale turbulence structures.

#### Acknowledgements

The authors acknowledge the valuable support of Lars Tretau of BAW Hamburg during the testing, setup and dune installation for the flume experiments in this study. Alexander Alfano of the Faculty of Geoscience, University of Bremen and MARUM is also acknowledged for proofreading the original draft.

#### **Author contributions**

Kevin Bobiles: Conceptualization, Data curation, Methodology, Validation, Investigation, Visualization, Formal analysis,

Writing – original draft, Writing – review & editing

Bernhard Kondziella: Methodology, Validation, Investigation, Writing – review & editing

Chrisitina Carstensen: Methodology, Validation, Investigation, Writing – review & editing

Ingrid Holzwarth: Conceptualization, Investigation, Writing – review & editing

Elda Miramontes: Investigation, Writing – review & editing

Alice Lefebvre: Conceptualization, Supervision, Methodology, Investigation, Writing - review & editing, Funding

acquisition

## 520 Data availability

The experimental data used in this study will be made available at https://www.pangaea.de/

# **Competing interests**

There are no competing interests among the authors.

## **Disclaimer**

Any opinions, findings and conclusions or recommendations expressed in this material are those of the author(s) and do not necessarily express the views of the institutions to where they belong.

#### **Financial support**

Kevin Bobiles and Alice Lefebvre have been funded for this study through the German Research Foundation (DFG) project FlowDEB 47010786.

#### 530 Review statement

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
