# Peer review of "Experimental Study of Time-averaged Flow and Turbulence Structures over Low-Angle Tidal Dunes under Steady Bidirectional Flows"

_EGUsphere, 2025_

## Author Comment (AC1)

Response to reviewers.

We would like to sincerely thank both reviewers for their constructive comments. We have answered them below.

Our comments are highlighted in blue. New text from the manuscript is highlighted in yellow.

**Reviewer 1**

This manuscript presents a high-quality and original study that successfully fills a relevant gap in the current research field. The experimental design is robust, and the data analysis is thorough and convincing. Overall, the scientific contribution is significant. Most of my feedback therefore concerns aspects of writing style, structure, and clarity rather than the scientific content itself. These revisions would enhance readability and ensure that the strength of the work is fully reflected in the text. Below I highlight major concerns, while in the attached pdf I added in-line questions and suggestions.

**Title:** The title currently includes the phrase *"Steady bidirectional flow"*, which is conceptually problematic, since a bidirectional flow, which fluctuates between +u_max and –u_max, cannot be steady by definition (as du/dt≠0). Upon reading further, it becomes clear that the authors refer to two separate experimental settings: a steady flow in one direction, and a steady flow in the opposite direction. While this design is valid, the term "steady bidirectional flow" may lead to initial confusion. I therefore recommend removing the word *"steady"* or rephrasing the title for improved clarity.

We have removed the mention of steady bidirectional flow in the title entirely to avoid confusion. The title is now: Experimental Study of Time-averaged Flow and Turbulence over Asymmetric Tidal Dunes under Bidirectional Flows.

**Abstract:** The abstract contains all relevant information, but its organization could be improved. The current structure appears somewhat disordered, which obscures the central message of the paper. A clearer and more conventional scientific structure is recommended. The sections that I miss or would improve include:

- Motivation: Briefly state why the research topic is important and what knowledge gap is being addressed.

We have provided the motivation of the study by stating in the first sentence of the abstract that asymmetric tidal dunes which are mainly intermediate- to low-angle dunes are the common types of dunes found in natural environments. We proceeded to state the knowledge gap by stating in the second sentence that these types of dunes have rarely received detailed in the past since most were focused on high-angle dunes which are typical only in the laboratory settings or large rivers and not in tide-dominated environments.

- Research question/objective: Clearly formulate the central question or hypothesis.

We have clearly provided the central questions or hypotheses of our study by stating in the abstract our two main research questions. It has been written as "Specifically, we aim to address how tidal dune shape, especially the lee side geometry, controls the properties

of flow separation and resulting turbulence structures. Furthermore, we address how flow bidirectionality changes flow and turbulence over the same tidal dune geometry."

- Methods: this could be slightly less detailed.

We have toned down the details of the method and results by restructuring lines 11 - 26 into a more compact descriptions of the methods and findings.

We have rewritten it as follows: "To achieve this, we conducted large-scale, high-resolution flume experiments over two idealised dune morphologies which represent natural asymmetric tidal dunes with intermediate- to low-angle slopes. The flow condition was an idealised representation of tidal flow for which the same unidirectional steady currents were imposed first in one direction, then in the opposite direction. Our results show that for the case of intermediate-angle tidal dune and when the flow was directed from the gentle stoss to the steep lee slope, a downward expanding turbulent wake and a small, near-bed permanent flow separation were detected. A small flow separation was also detected for the case of low-angle tidal dune. When the flow was reversed and directed from the steep stoss to the gentle lee slope, flow direction significantly altered the flow dynamics for both dunes as no permanent flow separation was observed and turbulence structure was similar to that over a flat bed. Interestingly, we demonstrated that a small intermittent flow separation can still form even for tidal dunes with very gentle slope ($4°$) provided that a short steep portion is present."

- Broader implications: End with the wider significance or applications of the findings.

We have added broader implication in the last sentence of the abstract by stating that our findings on the characteristics of flow separation and turbulence structures above tidal dunes can have implications on the parameterisations of hydraulic roughness, estimation of sediment transport and the resulting morphodynamics in natural shallow water environments.

"Overall, our study highlights the significant impact of dune morphology, particularly the lee side slopes, and flow direction on the flow and turbulence dynamics above asymmetric tidal dunes. Our findings can have further implications on the parameterisation of hydraulic roughness, estimation of sediment transport and the resulting morphodynamics in natural shallow water environments."

The main findings—especially regarding non-steep dunes, irregular dune shapes, and the importance of leeside angle and steep-face morphology—should be emphasized more clearly. At present, the abstract does not adequately convey the strength and clarity of the paper's contributions.

We have emphasized the main findings especially for non-steep irregular dune shape by delineating explicitly the findings for each type of dunes considered (intermediate- and low-angle tidal dunes). We have also stressed our findings when the flow is reversed since one of the main themes of our study was flow bidirectionality.

We have incorporated these by writing in the abstract as "A small flow separation was also detected for the case of low-angle tidal dune. When the flow was reversed and directed from the steep stoss to the gentle lee slope, flow direction significantly altered the flow dynamics for both dunes as no permanent flow separation was observed and turbulence structure was similar to that over a flat bed. Interestingly, we demonstrated that a small intermittent flow separation can still form even for tidal dunes with very gentle slope (4°) provided that a short steep portion is present. This implies that low-angle dunes can generate flow resistance and can potentially contribute to sediment mobilisation above low-angle dunes. Overall, our study highlights the significant impact of dune morphology, particularly the lee side slopes, and flow direction on the flow and turbulence dynamics above asymmetric tidal dunes."

**Introduction:** The introduction reads somewhat unstructured and would benefit from clearer logical progression and stronger narrative coherence. The reader needs more guidance throughout the section: Why is each paragraph or subtopic relevant? How does each section build toward the research question? What is the main message or takeaway from each part?

We have reorganised the Introduction to incorporate the suggestions so that the logical flow would become clearer. To follow a more consistent structure in the Introduction and to provide guidance to the readers at each section, the first paragraph provided the background and significance of the study. We have answered why this study is important in this paragraph. Second to fifth paragraphs were the current state of knowledge about dunes. Here, we have described what is currently known and unknown. Specifically, these paragraphs were organised into three subtopics. Second and third paragraphs focused on dune morphology, third to fifth paragraphs focused on flow dynamics above dunes and fifth paragraph focused on the importance of accounting for flow bidirectionality in the study of flow properties of dunes under tidal flows. We have provided our research questions or problem statements based from the preceding paragraphs in the sixth paragraph. The last paragraph explained how we are going to answer our research questions and we have also provided an outlook on the structure of the succeeding sections of this study.

In the first paragraph, we have described already the significance of the study and what is the current knowledge gap this study is trying to address. This way the reader would already have an idea what is the study all about and would have set the stage on what subtopics the succeeding paragraphs are going to introduce (i.e., dune morphology and flow dynamics above dunes. Lines 34 – 39 were rewritten into "Despite their prevalence in natural flow environments, the detailed characteristics of flow and turbulence structures over intermediate- to low-angle tidal dunes under unsteady, reversing tidal flows remain poorly studied, even though such conditions dominate many estuaries and tidal rivers. This knowledge gap in dune flow dynamics is necessary to address since dunes are one of the drivers of flow resistance and sediment transport in tidal rivers and coastal estuarine environments (Best, 2005; Coleman and Nikora, 2011; Venditti, 2013; De Lange et al., 2021). Resolving dune-induced flow and turbulence under tidal forcing is essential for understanding of hydro-morphodynamic feedbacks (Villard and Kostaschuk, 1998; Kostaschuk et al., 2004; Herrling et al., 2021) and for applications such as navigation and waterways management (Nasner et al., 2009) as well as subsea cable burial and offshore constructions."

In lines 41-47 of the second paragraph, to provide an explanation why this paragraph or subtopic is relevant, we have added an introductory sentence saying "Insights on the morphology of the dune being considered is valuable in the understanding of the flow and

turbulence dynamics." Second paragraph mainly introduced the intermediate- and low-angle dunes which are the two main dune morphologies considered in this study.

Lines 71-76 and lines 80-85 were merged and have become third paragraph of the Introduction to be consistent with the logical flow because the preceding paragraph was already introducing the concept of dune morphology and the types of dune morphology based from its mean lee slope. Here in the new third paragraph, we have further discussed how dune morphology, specifically its steep face, controls flow separation and the associated turbulence structures above dunes. We have also described the flow dynamics over dunes. Specifically, we first introduced the flow dynamics over high-angle dunes since these types of dunes are the most commonly studied in previous literatures and this provided a background for the intermediate- and low-angle tidal dunes which is the main focus of this study. We have further described in the fifth paragraph the contrasting shape between river and tidal dunes and stated that those tidal dunes have still received less attention in previous studies.

We have written the third paragraph as "Dune morphology influence the flow and turbulence structures primarily through the lee side geometry (Kwoll et al., 2016). In particular, the steep face, which is the lee side slope downstream of the crest that is steeper than the mean lee slope and its adjacent segments, strongly governs whether flow separation is permanent, intermittent or absent. When flow separation forms, the steep face controls the shape, extent and intensity of the flow separation and the resulting turbulent wake (Fig. 1) (Lefebvre et al., 2016; Lefebvre and Cisneros, 2023). The overall characteristics of the mean flow and turbulence structures over angle-of-repose dunes are already well documented (Nelson et al., 1993; Mclean et al., 1994; Bennett and Best, 1995; Venditti and Bennett, 2000; Best, 2005; Lefebvre et al., 2014). Above angle-of-repose dunes (Fig. 1a), flow accelerates over the stoss side until it reaches the crest and decelerates over the lee side forming a permanent flow separation zone where a reverse flow is observed. A shear layer, which separates the flow recirculation cell from the overlying undisturbed flow, forms and expands. Kelvin-Helmholtz instabilities develop along this separated shear layer where strong velocity gradient become unstable, generating periodic roll-up and shedding of vortices that give rise to a large turbulent wake. This turbulent wake expands upward and advects over the stoss side of the adjacent dune. Below this turbulent wake, a newly formed internal boundary layer develops, with a logarithmic velocity profile. Because of the resulting flow structure, a maximum horizontal velocity located at the crest develops and is expected to generate high bottom shear stress capable of generating bedload and suspended sediment transports contributing to the morphodynamic changes of bottom topography. Over high-angle dunes (Figure 1b), a flow separation and turbulent wake are found, but their size and intensity are reduced compared to those above angle-of-repose dunes (Lefebvre and Cisneros, 2023, Kwoll et al., 2016; Kwoll et al., 2017)."

We have further elaborated our point on the flow dynamics over intermediate- and low-angle tidal dunes in the fourth paragraph. For both these types of dunes, previous studies have speculated only either an intermittent flow separation or nonexistent and in this study we explicitly demonstrated what kind of flow separation would occur over these dunes with idealised representation of natural tidal dunes.

We described in the fifth paragraph the importance of accounting for the flow bidirectionality in the study of intermediate- and low-angle dunes because these types of dunes are mostly under tidal forcing in nature. We have written it as as "In tidal settings, not only the bedform crest shape, but also the reversing of the tidal currents have an influence on the interaction

between dune and flow. Most laboratory and numerical studies have focused on steady unidirectional currents whereas realistic tidal environments impose systematic flow reversal and evolving flow intensity over a tidal cycle. Recent flume experiments further emphasized the need to elucidate flow dynamics over estuarine and tidal dunes under conditions that better approximate tidal forcing (Carstensen and Holzwarth, 2023; Porcile et al., 2025). Flow bidirectionality changes which dune side acts as the effective lee side, potentially suppressing flow separation, modifying shear layer development, effective hydraulic roughness (Herrling et al., 2021; De Lange et al., 2021; De Lange et al., 2024) and reorganising turbulent events that drive momentum exchange and sediment suspension."

The sixth paragraph explicitly stated the research questions or problem statements of our study. We have added a new paragraph as sixth paragraph: "The present study, therefore, addresses the following research questions. How do intermediate- and low-angle tidal dune morphologies, specifically the mean lee slopes and steep face, control the presence and structure of flow separation and resulting turbulence? And how does flow bidirectionality, representative of natural ebb-flood flow reversals, reorganise flow separation and the associated turbulence structures over the same fixed dune geometry?"

We closed the Introduction by providing how we are going to address our research questions and a brief outlook on the content of the succeeding sections of our study. We have written the seventh paragraph as "To answer these questions, we conduct high resolution, large-scale flume experiments to measure time-averaged flow and turbulence above representative two-dimensional fixed tidal dune fields under idealised bidirectional flows. The laboratory dunes cover the intermediate- to low-angle configurations (< 17°) with segmented lee sides and steepest slopes positioned near the crest, consistent with tidal dune morphology (Fig. 2). The chosen slopes cover the expected transition from nonexistent to intermittent to permanent flow separation. The large-scale nature of our flume experiments enables robust measurements in the near bottom region of the flow and the imposed bidirectionality provides controlled insight into how flow reversal modifies dune-induced flow separation and turbulence. We first proceed by describing the experimental methodology of our flume experiments and our laboratory dunes. We then present the time-averaged flow field and other derived flow and turbulence structures for each dune and flow configuration. We proceed further with the interpretation of how dune morphology and flow bidirectionality interact in shaping the observed flow dynamics. Finally, we discuss some implications for realistic tidal dune dynamics and sediment transport processes and summarizes the key findings in this study."

Additionally, the introduction would be strengthened by including more recent literature, since many references are from 20 years ago, and much progress has been made in the field since then.

We have added additional recent literatures in the Introduction.

In lines 32-33, we added recent literatures (in bold face): "bedforms which are particularly abundant in lower reaches of rivers (Lange et al., 2008; **Cisneros et al., 2020**), estuaries (Aliotta and Perillo, 1987; Bradley et al., 2013; **De Lange et al., 2024)** and in coastal tidal environments **(Damen et al., 2018**), where they develop into large fields with complex morphologies."

In lines 49-51, we have rewritten it as "The overall characteristics of the mean flow and turbulence structures over high-angle dunes are already well documented (Nelson et al., 1993; Mclean et al., 1994; Bennett and Best, 1995; Venditti and Bennett, 2000; Best, 2005; **Lefebvre et al., 2014**)."

In lines 61-63, we have added a preceding sentence and was written as "a flow separation and turbulent wake are found, but their size and intensity are reduced compared to those above angle-of-repose dunes (**Lefebvre and Cisneros, 2023, Kwoll et al., 2016; Kwoll et al., 2017)**"

In lines 71-72, we have rewritten it as "Dune morphology influence the flow and turbulence structures primarily through the lee side geometry (**Kwoll et al., 2016**)."

In lines 87-89, we have rewritten it as "In tidal settings, not only the bedform crest shape, but also the reversing of the tidal currents have an influence on the interaction between dune and flow. Most laboratory and numerical studies have focused on steady unidirectional currents whereas realistic tidal environments impose systematic flow reversal and evolving flow intensity over a tidal cycle. Recent flume experiments further emphasized the need to elucidate flow dynamics over estuarine and tidal dunes under conditions that better approximate tidal forcing (**Carstensen and Holzwarth, 2023; Porcile et al., 2025**). Flow bidirectionality changes which dune side acts as the effective lee side, potentially suppressing flow separation, modifying shear layer development, effective hydraulic roughness (**Herrling et al., 2021; De Lange et al., 2021; De Lange et al., 2024**) and reorganising turbulent events that drive momentum exchange and sediment suspension."

**Methods and results:** These sections are excellent. The analysis is detailed, systematic, and well-executed. The figures and interpretations convincingly support the conclusions. The only thing I would appreciate is a bit more explanation on the quandrant analysis and the interpretation of these results.

Thank you for the comment.

Regarding the quadrant analysis, we have provided a visual representation of these events. We have added Figure 7 for this purpose.

[Figure]

**Figure 7. Conceptual diagram showing the different quadrant events used to characterise turbulent events using quadrant analysis.**

In Section 3.3 of Result section, we have further provided a simple explanation or description of these events for reference on the interpretation of the results. We have added a paragraph for this after line 311: "Quadrant 1 (Q1, outward interaction) events are fast water burst that moves upward and quadrant 3 (Q3, wallward, inward interaction) events are those slow water bursts that move downward. Both quadrants 1 and 3 are considered negative contributors to Reynolds stresses meaning that they contribute energy to the mean flow by extracting energy from turbulence such as shear layer vortices and coherent flow structures. Quadrant 2 (Q2, ejection) events are low-momentum near-bed fluid being thrown-up into the flow and quadrant 4 (Q4, sweep events) are those high-momentum fluid from above sweeping down into the flow. Both quadrants 2 and 4 are considered positive contributors to Reynolds stresses. Conversely, these turbulent events extract energy from the mean flow contributing to turbulence production."

**Discussion:** The discussion is generally well written and organised. It could be expanded slightly to answer three of my remaining questions.

Thank you for the comment. We have devoted a separate subsection for this. It is titled as 4.5 Further implications on hydraulic roughness, superimposed dunes and tidal flows.

Firstly, it would be valuable to include a short section on implications for the field, addressing how these findings might influence or refine conclusions from previous field studies (e.g., de Lange et al, 2021 (https://doi.org/10.1029/2021WR030329), Prokocki et al., 2022 (https://doi.org/10.1002/ esp.5364),  de Lange et al. 2024 (https://doi. org/10.1029/2023JF007340), and others).

We have addressed this question in the new section 4.5 and incorporating also the additional references. We have written it as "Finally, the present findings complement and help refine conclusions from previous field studies of dune and its associated hydraulic roughness. De Lange et al. (2021) showed that conventional dune geometry predictors underpredict the

spatial variability of hydraulic roughness in rivers and that their attempt to correlate roughness with dune lee slopes was not satisfactory. Our experimental results suggest that even dunes with modest lee slopes produce flow separation and macroturbulent structures which might not be captured by simple dune geometry roughness predictors. These other factors could be the intermittent features (intermittent flow separation and shear layer fluctuations) and flow phase dependency which might introduce roughness variability that is not apparent from dune morphology alone, explaining why lee slope metrics do not fully relate to roughness in field settings. Our controlled experiments which use idealised representation of natural tidal dunes validate previous conceptual framework on distinct regimes of dune morphology in a fluvial-tidal riverine setting (Prokocki et al., 2022), that in the tidal section of the river, dunes are mainly low-angle (ca. 10 - 15° lee slope) and generate only small flow separation with weaker wakes than high-angle dunes. In summary, integrating our experimental results with field studies on tidal dunes (De Lange et al., 2021; Prokocki et al., 2022; De Lange et al., 2024) underscores that even intermediate- to low angle dunes can still exert considerable impact to flow resistance. Their associated flow separation, if any, and turbulence characteristics must be accounted for to accurately predict hydraulic roughness and sedimental transport under realistic, unsteady tidal flow conditions."

Moreover, since the experimental design uses two opposing steady flows to approximate tidal dynamics, it would be useful to reflect on how an actual tidal flow—characterized by gradual acceleration and deceleration—might affect the observed results.

We have addressed this question in the new section 4.5. We have written it as "Another key consideration in natural tidal flows is that they are unsteady, increasing and decreasing during each tidal phase. This unsteadiness may modulate our observed flow and turbulence structures within the tidal cycle. In natural tidal flows, the continually changing flow velocity and direction mean that flow separation and turbulence structure do not have much time to establish a fully developed steady state. Although our flow condition in the experiment is strictly steady unidirectional in two opposite directions which is an idealisation of the real tidal dynamics, we have effectively provided, at a particular time in the tidal cycle, an instantaneous snapshot of the flow and turbulence structures above our intermediate- and low-angle dunes which can serve as a guidance or reference for interpretation of the flow dynamics over natural tidal dunes under realistic tidal flows."

Finally, how would superimposed dunes affect your findings? Could you just add one or two sentence speculating on this? The reason I'm suggesting this is that superimposed dunes are recently found to be very important for sediment transport (I refer to the work of Judith Zomer), and clearly superimposed dunes impact the leeside and steep side angle of the primary dune.

We have addressed this question in the new section 4.5. We have written it as "Our tested dune configurations consist only of one scale of dune made of straight lines, and no superimposed secondary dunes were considered. Superimposed dunes, which are small bedforms that ride on the primary dunes, would likely modulate the flow separation and turbulence structures and alter the sediment transport dynamics above intermediate-angle and low-angle tidal dunes. Previous studies have shown how the steepness of the primary dune lee side controls the presence of secondary dunes over the lee side of the primary dune (Zomer et al., 2021). For our intermediate-angle dune when the flow is directed from the

gentle stoss to the steep lee slope (DUNE1_F1), superimposed dunes cannot propagate further downstream owing to the steeper lee slope making the flow separation above the primary dune unaltered. For our low-angle dune under the same F1 flow (DUNE2_F1), the gentle lee slope (10°) allows the secondary dunes to propagate over the lee side which might break up the flow expansion zone and can suppress the already small flow separation forming above the primary dune (Dalrymple and Rhodes, 1995; Prokocki et al., 2022). From these two speculations on our tested dunes, it is clear that presence of secondary bedforms can effectively modify the primary dune effective lee side slope. Furthermore, these secondary dunes can also introduce their own micro-scale flow separation and turbulence which may collectively increase the total roughness (Zomer and Hoitink, 2024; Liu et al., 2025). Overall, the presence of superimposed dunes would induce additional form roughness and can either attenuate or enhance the primary dune flow separation and turbulence structures. This, in turn, would also influence turbulence and sediment flux over the primary dunes (Zomer and Hoitink, 2024)."

Citation: https://doi.org/10.5194/egusphere-2025-4883-RC1

**Detailed, inline questions and suggestions from Reviewer 1:**

I do not think that a bidirectional flow can be steady. A steady flow is defined by dv/dt = 0. A bidirectional flow (representing tides) has a time component, so this is not steady.

We have removed the mention of "steady" in the title entirely to avoid confusion. The title is now: Experimental Study of Time-averaged Flow and Turbulence over Asymmetric Tidal Dunes under Bidirectional Flows.

Is it possible to start the abstract with an introduction and problem statement, instead of your methods? Why do we want to investigate flow over a dune?

In my opinion, it makes more sense to follow the general stucture of a paper, and give the reader an idea why you conducted this study and why they are reading this in the first place.

Thank you for this very insightful suggestion. I agree with your suggestion to follow a general structure for the abstract. Since the comment will end up reorganising the content, I rewrote the entire abstract incorporating all your individual comments in this section. The revised abstract has been rewritten as "Asymmetric tidal dunes with intermediate (10-17°) to low-angle slopes (< 17°), usually with an irregularly-shaped lee side, are often found in natural, constrained tidal environments such as tidal rivers, estuaries and tidal channels. However, previous studies on bedform flow dynamics have largely focused on high-angle dunes with a simple (straight) lee side, generally found in flume studies or small rivers. This study provides a detailed characterisation of the flow and turbulence over asymmetric tidal dunes under an idealised tidal flow condition based on laboratory measurements. Specifically, we aim to address how tidal dune shape, especially the lee side geometry, controls the properties of flow separation and resulting turbulence structures. Furthermore, we address how flow bidirectionality changes flow and turbulence over the same tidal dune geometry. To achieve this, we conducted large-scale, high-resolution flume experiments over two idealised dune morphologies which represent natural asymmetric tidal dunes with intermediate- to low-angle slopes. The flow condition was an idealised representation of tidal

flow for which the same unidirectional steady currents were imposed first in one direction, then in the opposite direction. Our results show that for the case of intermediate-angle tidal dune and when the flow was directed from the gentle stoss to the steep lee slope, a downward expanding turbulent wake and a small, near-bed permanent flow separation were detected. A small flow separation was also detected for the case of low-angle tidal dune. When the flow was reversed and directed from the steep stoss to the gentle lee slope, flow direction significantly altered the flow dynamics for both dunes as no permanent flow separation was observed and turbulence structure was similar to that over a flat bed. Interestingly, we demonstrated that a small intermittent flow separation can still form even for tidal dunes with very gentle slope (4°) provided that a short steep portion is present. This implies that low-angle dunes can generate flow resistance and can potentially contribute to sediment mobilisation above low-angle dunes. Overall, our study highlights the significant impact of dune morphology, particularly the lee side slopes, and flow direction on the flow and turbulence dynamics above asymmetric tidal dunes. Our findings can have further implications on the parameterisation of hydraulic roughness, estimation of sediment transport and the resulting morphodynamics in natural shallow water environments."

Lines 13-14: This is very detailed and possibly too long. I think first the study aim needs to be given (the sentence after these highlighted sentences), before you dive into the specifics of the study setup.

Suggestion for making it shorter (still need to reshuffle the order though): Two idealised dune morphologies were used to represent natural asymmetric tidal dunes with low to intermediate-angle slopes.

We have rewritten it as "To achieve this, we conducted large-scale, high-resolution flume experiments over two idealised dune morphologies which represent natural asymmetric tidal dunes with intermediate- to low-angle slopes."

Line 16: again, this can be more consise. Before you wrote how the dune morphologies are a representation of tidal dunes, now you say how the flow is a representation of tidal flow. You can make this easier to read by simply indicating that you are imitating the tidal environment.

We have rewritten it as "The flow condition was an idealised representation of tidal flow for which the same unidirectional steady currents were imposed first in one direction, then in the opposite direction."

Lines 16 -17: This is nothing new, right? We know for a while already that the flow seperation over steeper dunes is larger than over less steep dunes. If so, I think this sentence does not add much to your abstract. Or do I interpret this incorrectly?

We rephrased this to focus more on the features we observed for our intermediate-angle and low-angle tidal dunes instead of comparing our results to that of high-angle dunes. We have rewritten it as "Our results show that for the case of intermediate-angle tidal dune and when the flow was directed from the gentle stoss to the steep lee slope, a downward expanding turbulent wake and a small, near-bed permanent flow separation were detected. A small flow separation was also detected for the case of low-angle tidal dune."

Line 18: maybe stay away from complicated vocabulary in the abstract, or at least give a definition. Feel free to use it in the main body of the paper though!

We just deleted this as it is now irrelevant to the revised Abstract.

Line 19: of the dune at angle of repose, or the intermediate angle dune?

We referred to the turbulent wake that forms above the intermediate-angle dune when the flow is directed from the gentle stoss to the steep lee side. We have made it clear by rewriting as "Our results show that for the case of intermediate-angle tidal dune and when the flow was directed from the gentle stoss to the steep lee slope, a downward expanding turbulent wake and a small, near-bed permanent flow separation were detected."

Line 20: plural?

We have rewritten it to be clear. It is rewritten as "Small flow separation was also detected for the case of low-angle tidal dune."

Lines 21-22: I feel like this sentence is the novelty of this research! Where dunes are not simply idealized as triangles, but having a lee side with a steep side - as shown by Alice Lefebvres work earlier. It's awesome you can show this in the lab now! However, I feel like that this main finding (or at least, what I consider the main finding) should be stressed more, while other, more trivial and redundant information, can be removed from the abstract. Why not introduce the abstract/paper with the notion of "irregular" leeside angles? (I'm sure you have better terms for this than me).

Thank you for the comment. We highlighted this finding by putting it in a separate sentence and describing it in more general way without the specific details of the slopes to reduce the information in the abstract. We have rewritten it as "Interestingly, we demonstrated that a small intermittent flow separation can still form even for tidal dunes with very gentle slope (4°) provided that a short steep portion is present. This implies that low-angle dunes can generate flow resistance and can potentially contribute to sediment mobilisation above low-angle dunes."

Line 22: are we still talking about the leeside, or does this already relate to the next sentence which describes flow over the stoss side.

We are referring to the case of intermediate angle dune when the flow is directed from the steep stoss to the gentle lee slope. We have rewritten it clearly as "When the flow was reversed and directed from the steep stoss to the gentle lee slope, flow direction significantly altered the flow dynamics for both dunes as no permanent flow separation was observed and turbulence structure was similar to that over a flat bed."

Line 22: I don't think you can pluralize it like this.

We have now corrected it and rewritten this sentence as "When the flow was reversed and directed from the steep stoss to the gentle lee slope, flow direction significantly altered the flow dynamics for both dunes as no permanent flow separation was observed and turbulence structure was similar to that over a flat bed."

Line 23: nor?

We rephrased this sentence and just rewrote as "When the flow was reversed and directed from the steep stoss to the gentle lee slope, flow direction significantly altered the flow dynamics for both dunes as no permanent flow separation was observed and turbulence structure was similar to that over a flat bed."

Lines 26-27: Cool! But I think it needs an additional sentence on the implications of this research. Why is this important? What does this add in terms of scientific progress?

We have added a last sentence after Line 27 for this and wrote as "Our findings can have further implications on the parameterisation of hydraulic roughness, estimation of sediment transport and the resulting morphodynamics in natural shallow water environments."

Line 32: Dunes do not necessarily need to be sandy, and can also form in silt and gravel. You are right that they are mostly in sand-bedded environments, but to claim this in their definition goes a bit too far.

Thank you for pointing it out. We just removed this word to avoid too much generalisation. We rewrote this sentence as "Among these, dunes are large (decimeters to meters in height, meters to hundreds of meters in length) bedforms which are particularly abundant in lower reaches of rivers (Lange et al., 2008; Cisneros et al., 2020), estuaries (Aliotta and Perillo, 1987; Bradley et al., 2013; De Lange et al., 2024) and in coastal tidal environments (Damen et al., 2018), where they develop into large fields with complex morphologies."

Line 32: Maybe cite the newer paper of Cicneros as well?
https://www.nature.com/articles/s41561-019-0511-7

We included now the paper of Cisneros et al. (2020). Similarly, we have rewritten this sentence as "Among these, dunes are large (decimeters to meters in height, meters to hundreds of meters in length) bedforms which are particularly abundant in lower reaches of rivers (Lange et al., 2008; **Cisneros et al., 2020**), estuaries (Aliotta and Perillo, 1987; Bradley et al., 2013; De Lange et al., 2024) and in coastal tidal environments (Damen et al., 2018), where they develop into large fields with complex morphologies."

Line 33: Maybe cite the newer paper of de Lange as well?

https://doi. org/10.1029/2023JF007340

We included now the paper of de Lange et al. (2024). Similarly, we have rewritten this sentence as "Among these, dunes are large (decimeters to meters in height, meters to hundreds of meters in length) bedforms which are particularly abundant in lower reaches of rivers (Lange et al., 2008; **Cisneros et al., 2020**), estuaries (Aliotta and Perillo, 1987; Bradley et al., 2013; De Lange et al., 2024) and in coastal tidal environments (Damen et al., 2018), where they develop into large fields with complex morphologies."

Lines 34-35: not entirely true, check https://doi.org/10.1029/2021WR030329

We rephrased this sentence to say as "This knowledge gap in dune flow dynamics is necessary to address since dunes are one of the drivers of flow resistance and sediment transport in tidal rivers and coastal estuarine environments (Best, 2005; Coleman and Nikora, 2011; Venditti, 2013; De Lange et al., 2021)."

Lines 42 – 47: I'm not entirely sure about this paragraph. Although I really like the subdivision you are making here, I feel like you pull a lot of information from different sources and combine them to make generalizations, while I'm not sure you can make those generalizations irght away. I would love it if you are right, but I think this requires proof with statistics about those different dunes and environments. I do not think any of the papers you cited do this (they all just offer pieces of the puzzle), therefore, this claim is too hard. It would honestly be worth proving these claims and writing a paper about this! Or this exists already, then I'm wrong, but then citing this would be great.

Thank you for the insightful comments about the classification. We agree that a more detailed analysis is necessary to establish the formal classification of dunes based on mean lee slopes. However, there is still no established criteria on classifying dunes on the basis of dune lee slope alone. Since it is important for this study to have some way of identifying our dunes based on mean lee slope, we based our classification from the recent study of Lefebvre and Cisneros (2023) where they proposed these delineations based from their numerical simulations (high-angle dune with permanent flow separation: mean lee slope > 17°, intermediate-angle dune with only intermittent flow separation: mean lee slope ca 10-17° and low-angle dune with nonexistent flow separation: mean lee slope < 10°). I also added an additional explanation at the end of this paragraph that this classification is not a strict classification but rather proposed one based from several factors.

From this, we have largely rewritten lines 41-47 as "Insights on the morphology of the dune being considered is valuable in the understanding of the flow and turbulence dynamics. Depending on their morphology, dunes can be classified as high-, intermediate- or low-angle dunes based on their mean lee slopes (Lefebvre and Cisneros, 2023) (Fig. 1). Angle-of-repose (slope > 24°) and high-angle dunes (slope > 17°) have heights usually on the order of 1/6 of the water depth (Bradley and Venditti, 2017) and are commonly found in small rivers and laboratory flumes (Venditti et al., 2005; Naqshband et al., 2014) where a unidirectional current is the dominant flow condition. Intermediate dunes have lee slopes between ca. 10 and 17° and are often found in large rivers (Cisneros et al., 2022). Low-angle dunes with lee slopes of less than 10° are mostly found in large tidal rivers, estuaries and tide-dominated shelves (Nasner, 1974; Aliotta and Perillo, 1987; Lefebvre et al., 2021). Importantly, however, these associations between dune lee slopes, environment type and main flow forcing should be viewed as tendencies rather than strict classifications because dune classification reflects the combined influenced of several factors such as dune morphology, environment, hydrodynamics, sedimentology and migration dynamics to name a few."

Line 54: maybe explain this?

We rewrote this sentence to define explicitly Kelvin-Helmholtz instabilities as "Kelvin-Helmholtz instabilities develop along this separated shear layer where strong velocity gradient become unstable, generating periodic roll-up and shedding of vortices that give rise to a large turbulent wake."

Line 62: but/however

We have deleted this and reformed the sentence as "Intermediate-angle dunes (Figure 1c) rarely have a permanent flow separation but are likely to possess an intermittent flow

separation (Kostaschuk and Villard, 1996; Roden, 1998; Best and Kostaschuk, 2002; Sukhodolov et al., 2006; Lefebvre and Cisneros, 2023)."

Lines 62-63: aha, so you're study is about flow structures on intermediate/low dunes. I would have loved it if this research gap would have become more clear in te abstract, and in the first section of your introduction, so that the reader knows why they are reading all of this information.

Thank you for pointing it out. I moved this information in the first paragraph of the Introduction to make it clear that this is the focus of my study. We have rewritten this main point of our study in the first paragraph as "Despite their prevalence in natural flow environments, the detailed characteristics of flow and turbulence structures over intermediate- to low-angle tidal dunes under unsteady, reversing tidal flows remain poorly studied, even though such conditions dominate many estuaries and tidal rivers. This knowledge gap in dune flow dynamics is necessary to address since dunes are one of the drivers of flow resistance and sediment transport in tidal rivers and coastal estuarine environments (Best, 2005; Coleman and Nikora, 2011; Venditti, 2013; De Lange et al., 2021)."

Line 63: I mean, they do show this via their modelling results, or not?

We have deleted this word because we reformed the sentence. We now have rewritten it as "Intermediate-angle dunes (Figure 1c) rarely have a permanent flow separation but are likely to possess an intermittent flow separation (Kostaschuk and Villard, 1996; Roden, 1998; Best and Kostaschuk, 2002; Sukhodolov et al., 2006; Lefebvre and Cisneros, 2023)"

Line 71: the entire morphology, or the angles of the sides, like you argued in the sections before?

We referred here to the slope (mean slope and steep slope) of the lee side of the dune.

Line 72: I feel like this can be organized a bit better. For example, this section could be combined with the section on line 40/45.

We have reorganised lines 71-76 and placed them in third paragraph of the introduction. This comes after lines 40-45 which is the second paragraph that focus on the classification of dune morphology. It is now written as "Dune morphology influence the flow and turbulence structures primarily through the lee side geometry (Kwoll et al., 2016). In particular, the steep face, which is the lee side slope downstream of the crest that is steeper than the mean lee slope and its adjacent segments, strongly governs whether flow separation is permanent, intermittent or absent. When flow separation forms, the steep face controls the shape, extent and intensity of the flow separation and the resulting turbulent wake (Fig. 1) (Lefebvre et al., 2016; Lefebvre and Cisneros, 2023). The overall characteristics of the mean flow and turbulence structures over angle-of-repose dunes are already well documented (Nelson et al., 1993; Mclean et al., 1994; Bennett and Best, 1995; Venditti and Bennett, 2000; Best, 2005; Lefebvre et al., 2014). Above angle-of-repose dunes (Fig. 1a), flow accelerates over the stoss side until it reaches the crest and decelerates over the lee side forming a permanent flow separation zone where a reverse flow is observed. A shear layer, which separates the flow recirculation cell from the overlying undisturbed flow, forms and expands. Kelvin-Helmholtz instabilities develop along this separated shear layer where strong velocity gradient become

unstable, generating periodic roll-up and shedding of vortices that give rise to a large turbulent wake. This turbulent wake expands upward and advects over the stoss side of the adjacent dune. Below this turbulent wake, a newly formed internal boundary layer develops, with a logarithmic velocity profile. Because of the resulting flow structure, a maximum horizontal velocity located at the crest develops and is expected to generate high bottom shear stress capable of generating bedload and suspended sediment transports contributing to the morphodynamic changes of bottom topography. Over high-angle dunes (Figure 1b), a flow separation and turbulent wake are found, but their size and intensity are reduced compared to those above angle-of-repose dunes (Lefebvre and Cisneros, 2023, Kwoll et al., 2016; Kwoll et al., 2017)"

Line 78: maybe additionally define TW in the caption. Also, could you add the degrees for the steep face of the intermediate and low-angle dune?

We have added the definition of TW in the Figure caption as shown below.

[Figure]

**Figure 1: Types of dunes based on their morphology. TW: Turbulent wake.**

Regarding the angles of the steep faces for the intermediate and low-angle dunes, I also thought of putting in the drawing. However, there is no clear range for the angle of steep face per type of dune. We have the angle of the steep face for our idealised dunes but I think this figure (Fig. 1) is more for general depiction of dunes so the angle of steep face is variable.

Line 80: Again, this section could be merged with the section on line 40/45. The flow of the introduction is quite lost by all these different sections not being interlinked.

We have also merged lines 80-85 with lines 71-76 because they both discussed about dune morphology and its influence on flow dynamics above dunes. We have also discussed here the flow dynamics over angle-of-repose dune as reference. It is rewritten as third paragraph of the introduction "Dune morphology influence the flow and turbulence structures primarily through the lee side geometry (Kwoll et al., 2016). In particular, the steep face, which is the lee side slope downstream of the crest that is steeper than the mean lee slope and its adjacent segments, strongly governs whether flow separation is permanent, intermittent or absent. When flow separation forms, the steep face controls the shape, extent and intensity of the flow separation and the resulting turbulent wake (Fig. 1) (Lefebvre et al., 2016; Lefebvre and Cisneros, 2023). The overall characteristics of the mean flow and turbulence structures over angle-of-repose dunes are already well documented (Nelson et al., 1993; Mclean et al., 1994; Bennett and Best, 1995; Venditti and Bennett, 2000; Best, 2005; Lefebvre et al., 2014). Above angle-of-repose dunes (Fig. 1a), flow accelerates over the stoss side until it reaches the crest and decelerates over the lee side forming a permanent flow separation zone where a reverse flow is observed. A shear layer, which separates the flow recirculation cell from the overlying undisturbed flow, forms and expands. Kelvin-Helmholtz instabilities develop along this separated shear layer where strong velocity gradient become unstable, generating periodic roll-up and shedding of vortices that give rise to a large turbulent wake. This turbulent wake expands upward and advects over the stoss side of the adjacent dune. Below this turbulent wake, a newly formed internal boundary layer develops, with a logarithmic velocity profile. Because of the resulting flow structure, a maximum horizontal velocity located at the crest develops and is expected to generate high bottom shear stress capable of generating bedload and suspended sediment transports contributing to the morphodynamic changes of bottom topography. Over high-angle dunes (Figure 1b), a flow separation and turbulent wake are found, but their size and intensity are reduced compared to those above angle-of-repose dunes (Lefebvre and Cisneros, 2023, Kwoll et al., 2016; Kwoll et al., 2017)"

Line 94: maybe use "laboratory dunes"? For me this feels like modelled almost always refers to a computer model.

We have changed the words into "laboratory dunes". It appeared in the last paragraph of the Introduction and written as "The laboratory dunes cover the intermediate- to low-angle configurations (< 17°) with segmented lee sides and steepest slopes positioned near the crest, consistent with tidal dune morphology (Fig. 2)."

Line 139: can you add the meaning of H, L, F2 and F1 in the caption?

We have added their definitions in the Figure caption as shown below.

[Figure]

**Figure 4. Schematic diagram of tidal dune morphology. H is the bedform height, L is the bedform length, F1 refers to the flow that is aligned with the dune asymmetry (i.e., the flow is directed from the gentle stoss to the steep lee slope) and F2 refers to the flow that is opposing the dune asymmetry (i.e., the flow is directed from the steep stoss to the gentle lee slope).**

Lines 143-145: aha! Now I understand what you mean with "steady" "bidiretional" flows. You are actually just representing one flow velocity, but in both directions. II think it might be nice to highlight this already earlier in the paper. However, I still struggle with using the term steady...

Thank you also for pointing it out. We now explicitly mentioned in the revised version this idealisation of our flow condition in the abstract as well. We take note your concern with the term "Steady" so I just simply dropped it for clarity.

In the abstract, we specifically highlighted this and is written as "The flow condition was an idealised representation of tidal flow for which the same unidirectional steady currents were imposed first in one direction, then in the opposite direction."

Line 150: Do you define those parameters somewhere?

We added a definition of this parameter in the preceeding sentence to add context. It is written as "The Froude number which is the ratio of the inertial force to the restoring gravitational force (Fr = U/√(gh)) provides the relative importance of inertial forces acting on fluid particles to the weight of the particle. The Froude number is a widely used parameter in scaling down field to laboratory properties involving free surface flow."

Line 157: Repetition of the second sentence of this section. I suggest merging.

We maintained this line and instead modified second sentence. The second sentence in lines 153-154 was written as "The fixed dunes were made from concrete and were fine-sandblasted to provide a natural grain roughness."

Line 158: How wide where the concrete dunes? Did they cover the entire width of the flume?

Each concrete dune has a width of 1.4 m. The flume width is 1.5 m. To facilitate deployment and installation of dunes into the flume, we provided small gaps at each side of the dune (5 cm each side).

Line 161: The F1 and F2 sign in the left figures confuse me. They make it seem like of the measurement region, the left half is measured under F2, while the right half is measured under F1-conditions. Consider removing, or at least not aligning it so perfectly with the measurement region.

I just placed them at both sides of each figure to avoid confusion. But I think it is still good to show them to provide visuals for the flow directions. The new Figure 5 is shown below

[Figure]

**Figure 5. Laboratory tidal dunes and experimental flume setup.**

Lines 175-176: I do not understand this. What do you mean, alighned/opposed with the dune assymetry?

We made this clear by following the suggestions from the second reviewer. F1 flow = flow is aligned with dune asymmetry = flow is directed from the gentle stoss to the steep lee slope. F2 flow = flow is opposed to the dune asymmetry = flow is directed from the steep stoss to the gentle lee slope. We have now adopted this convention for the entire manuscript.

Line 184: Can you elaborate on this?

We have elaborated the further quality control and wrote it as "After despiking the data (Goring and Nikora, 2002), further quality control was performed by inspection of the data signal strength such as the signal-to-noise ratio (SNR) and correlation. Bad quality data not filtered by the above despiking method such as those from multi-reflection coming from the bottom surface were discarded and replaced when their corresponding SNR and correlation were below 15 dB and 80%, respectively."

Lines 218-220: I find this a bit hard to grasp. Would it be possible to provide a visualisation of the quadrants and their meanings?

We have added a figure to provide visualisation to these quadrant events. It appeared as Figure 7 as shown below.

[Figure]

**Figure 7. Conceptual diagram showing the different quadrant events used to characterise turbulent events using quadrant analysis.**

Line 239: I think it would improve the readibility and main message of this figure, if the data is interpolated and shown as a surface plot instead.

We have replotted it as surface plots. It is shown as Figure 8 shown below

[Figure]

**Figure 8. Time-averaged flow fields. I) DUNE1_F1, II) DUNE1_F2, III) DUNE2_F1, IV) DUNE2_F2.**

Lines 271-273: I think it might be beneficial to the readability to zoom in, just like you did in Figure 9.

I enlarged the figure similar to Figure 9. It appeared now as Figure 11.

[Figure]

**Figure 11. Vertical gradient of time-averaged horizontal velocity, $\partial \overline{u}/\partial z$ (/s). a) DUNE1_F1, b) DUNE2_F1, c) DUNE1_F2, d) DUNE2_F2.**

Lines 296-297: I wonder if your results would align if ou would use the the law of the wall principle? Such as done in this work, see equation 12 https://doi.org/10.1029/2024WR037065 Or is this method not applicable to your work?

Thank you for this interesting comment. While the law of the wall can be also used to provide estimate of the bottom shear stress, we believed that they cannot be compared directly with the results obtained from the Reynold stress. The total bottom shear stress obtained here was based from the spatially-averaged Reynolds stresses. This means that the obtained bottom shear stress is an overall estimate along the dune (like an averaged total bottom shear stress for the entire dune) and not at a specific location along the dune. On the other hand, the law of the wall estimate the friction velocity (and from here the total bottom shear stress can be estimated) at a single velocity profile. So, it is an estimate of the bottom shear stress at a particular location and not for the entire dune.

Line 316: Could you give an explanation/description of the four different events? I find it hard to interpret section 3.3.

We have provided a description for the four different quadrant events at the first paragraph of this section. It was written as "Quadrant 1 (Q1, outward interaction) events are fast water burst that moves upward and quadrant 3 (Q3, wallward, inward interaction) events are those slow water bursts that move downward. Both quadrants 1 and 3 are considered negative contributors to Reynolds stresses meaning that they contribute energy to the mean flow by extracting energy from turbulence such as shear layer vortices and coherent flow structures. Quadrant 2 (Q2, ejection) events are low-momentum near-bed fluid being thrown-up into the flow and quadrant 4 (Q4, sweep events) are those high-momentum fluid from above sweeping down into the flow. Both quadrants 2 and 4 are considered positive contributors to Reynolds stresses. Conversely, these turbulent events extract energy from the mean flow contributing to turbulence production."

Line 367: I really like this diagram, well done! :)

Thank you for the comment

Lines 402-403: I would love to hear more about what these results mean in the field. Would it change conclusions from other studies

We have provided more explanations on the influence of flow bidirectionality as well as its implications in the field. We have written this after Line 403 as "Flow bidirectionality effectively switches the flow dynamics between a more separated flow regime (F1 flow) and an attached near-bed flow regime (F2 flow) implying that flow separation metrics such as flow separation intermittency and size do not solely depend on morphology but also on flow orientation relative to the dune asymmetry. In the field, this can imply that flow separation may depend strongly on the phase of the tidal cycle and the instantaneous flow separation metrics may not be generalised across the entire tide cycle. Furthermore, field implications highlight that bedforms and their associated flow and turbulence structures respond to changing forcing and can exhibit spatial and temporal variability, consistent with separation regimes that switch with flow reversal."

Lines 454-456: I'm not sure what you mean here. Could you tie this in with another section? Doing this avoids a section that consists of only 2 lines, and would give it more context, possibly clarifying what you mean.

We just reformed this paragraph and focus more on the influence of flow bidirectionality on the large-scale turbulence structure. This provides more context on the bidirectionality

concept being discussed in this paragraph. It was rewritten as "Moreover, flow bidirectionality can also exert considerable influence on the organisation and strength of large-scale turbulence structures. Shear layer development and turbulence production that promote spatially coherent, large-scale turbulence structures are mostly pronounced over intermediate-angle dune and when the flow is directed from the gentle stoss to the steep lee slope. These features are, however, effectively damped when the flow is directed from the steep stoss to the gentle lee slope with the strongest suppression for the case of low-angle dune. This directionally-dependent modulation, similar to that happening for flow separation and turbulent wake, demonstrates that, depending on flow direction, a dune can switch between a macroturbulence-active regime (when the flow goes from the gentle to the steep side, as in our F1 flow conditions) and a largely attached, weakly coherent structure (when the flow goes from the steep to the gentle side, as in our F2 flow conditions), highlighting the importance of accounting for flow reversal in the flow dynamics over tidal dunes."

Lines 467-468: true, with potentially pretty impactful consequences: https://doi.org/10.1038/s41467-025-61248-5

Thank you for the comment. We have added the additional reference and rewritten as "for the observed elevated suspended sediment concentration on both the crest and lee side of the dune (Thorne et al., 1989; Kostaschuk and Church, 1993; De Lange et al., 2025)."

Line 480: could you give number here? What does "almost" mean? (maybe in percentage flow depth).

Our word "almost" was a bit confusing since flow separation is always attached to the bed. We changed the highlighted words into "thin" to refer to the thickness of the flow separation and provided its thickness as percentage of flow depth (14% of flow depth). We have reformed the sentence as "2. Over the intermediate-angle dune, the elongated permanent flow separation is short and thin (14% of the flow depth)."

Lines 495-496: What are the concequences of this?

We have added a sentence explaining the importance of large-scale turbulence to sediment transport and mixing processes in natural flows. It was written as "Identification and accounting for large-scale turbulence structure in sediment transport are important as they act as principal drivers of sediment suspension and vertical mixing processes found in natural flow environments."

**Reviewer 2**

The manuscript titled **"Experimental Study of Time-averaged Flow and Turbulence Structures over Low-Angle Tidal Dunes under Steady Bidirectional Flows"** addresses an important knowledge gap in our understanding of flow structure over bedforms under bidirectional flow conditions. This paper uses physical flume experiments to examine how two fixedconcrete dunes, representing low- and intermediate-angle tidal dunes, scaled from the Weser Estuary, shape flow separation, turbulence, and bed shear stress under reversing flows. For each dune, steady flow is run in two directions representing ebb and flood flows, and velocity fields are measured with an ADV and interpolated to 2D sections over the dunes. From these, the authors derive mean flow, separation zones, turbulent kinetic energy, Reynolds stress, and the distribution of turbulent events.

In this manuscript, the authors highlight previously unknown flow characteristics over tidal bedforms. They show that even intermediate- and low-angle tidal dunes can sustain permanent flow separation during ebb flows, but that separation length and thickness shrink markedly as lee slope decreases. Bed shear stress and the frequency of energetic ejection and sweep events are maximized over the intermediate-angle dune with flow over the steep face. Flow reversal (i.e., flood flows) eliminates permanent flow separation and any coherent bed-attached wakes. The study concludes that lee-slope geometry and flow direction together control separation, macroturbulence, and effective hydraulic roughness of tidal dunes.

The manuscript presents new and interesting results, the and the figures are clear and well-labelled; however, the current results and discussion would benefit from reorganization and clearer, more consistent writing so that these novel findings are easier to follow and more effectively highlighted. The following are some suggested changes that the authors should consider during revisions:

1. It would be great if the authors could adopt systematic configuration labeling for the four flow configurations throughout the manuscript (e.g., DUNE1_F1, DUNE2_F2), rather than using phrases such as "here" or "in this case."For instance: "High TKE is concentrated within the immediate vicinity of the bed and diminishes towards the upper portion of the water column. Although not as strong as the previous case (Fig. 11a), high TKE occurs within the trough and the immediate downstream portion of the gentle side." — I assumed that the authors were discussing TKE for DUNE2_F1. However, this is not very clear because, even though they say, "although not as strong as the previous case," they did not discuss TKE for DUNE1_F1 (which I assume is the previous case) immediately beforehand.

   Thank you for the helpful suggestion. Your suggestion makes the presentation of results more systematic and easier to follow. We have adopted and sticked to the following configurations (i.e., DUNE1_F1, DUNE1_F2, DUNE2_F1 and DUNE2_F2) throughout the Results section. We have also used the same configurations whenever there is a need to refer to a particular configuration in the Discussion section.

2. In some parts of the Results and Discussion sections, the authors use phrases such as "flow over the steep slope" and "flow over the gentle slope," etc. I suggest changing these to "flow directed from the gentle to the steep slope" and "flow directed from the steep to the gentle slope," or using F1/F2 within parentheses. While I understand that "flow over the steep slope" would represent the F1 condition, this is confusing because

the flow would technically pass over both the steep and gentle slopes regardless of flow direction.

Thank you for the valuable suggestions. This greatly improves the consistency when referring to F1 and F2 flows. We have now consistently adapted this labelling throughout the manuscript especially in the Result and Discussion sections.

The following conventions refer to the same flow conditions:

F1 flow = flow is aligned with the dune asymmetry = flow is directed from the gentle stoss to the steep lee slope

F2 flow = flow is opposed to the dune asymmetry = flow is directed from the steep stoss to the gentle lee slope

In the same vein, the authors should consider a consistent sequence within each Results subsection: for each metric (such as TKE or velocity), a configuration-by-configuration description, with paragraph breaks or clear indications when switching to a different flow direction. Given that the motivation is understanding tidal flows, each subsection should explicitly report how reversing flow (F1 to F2) alters each metric for each dune, rather than introducing bidirectionality in isolated passages that read as ancillary.

We have rewritten each Results subsection following this consistent way of describing the results. Furthermore, we explicitly described for each metric the F2 flow case to bring out the main theme of flow bidirectionality. They are now reflected accordingly in the revised manuscript following also the above first two suggestions.

Lines 231-235 have been rewritten as "The time-averaged flow fields for both dunes with F1 and F2 flow setups are shown in Fig. 8.

For DUNE1_F1, a strong deceleration zone characterised by slow mean streamwise velocity and downward directed mean vertical velocity is observed over the steep lee side of the dune. A gradually accelerating flow characterised by the increasing mean streamwise velocity and an upward directed mean vertical velocity is seen above the gentle stoss side of the dune.

For DUNE2_F1, the same flow pattern as that of DUNE1_F1 can be observed although its magnitude is decreased especially the mean streamwise velocity.

When the flow is reversed and is now directed from the steep stoss to the gentle lee slope (F2 flow), the acceleration-deceleration flow pattern reverses its occurrence above the dunes. For DUNE1_F2, a weaker deceleration zone compared to that of DUNE1_F1 is observed over the gentle lee side as evident from the gradually decreasing pattern of the mean streamwise velocity. As the stoss side is now steeper than the lee side, a strong upward mean vertical velocity is observed indicating the influence of flow reversal on the mean flow characteristics.

The same flow characteristics are also observed for DUNE2_F2 with some subtle features due to the presence of short steep section at the gentle lee side. At the location of the short steep section (ca x= 38.0 m), a stronger deceleration zone characterised by

slower mean streamwise velocity and stronger downward mean vertical velocity than the rest of the flow structure above the gentle lee slope of the dune is observed.

Overall, these observations confirm the influence of topographic forcing on the time-averaged flow fields."

Lines 241-247 have been rewritten as "A permanent flow separation zone can be observed very close to the dune surface for both dunes over their steep faces, when the flow is directed from the gentle stoss to the steep lee slope (F1 flow) (Fig. 9). Both flow separations start to develop at or shortly after the brink point, which marks the beginning of the steep face, and are extending over the steep lee side of the dune. Although both dunes show the presence of flow separation, their sizes are different. For DUNE1_F1, the flow separation length is $L_{FSZ}$ = 2.3H or 3.0$H_{SF}$. The maximum thickness is $Th_{FSZ}$ = 0.14H or 0.18$H_{SF}$. For DUNE2_F1, the flow separation length and thickness are much shorter and thinner than that of DUNE1_F1 with dimensions of $L_{FSZ}$ = 1.25H or 3.74$H_{SF}$ and $Th_{FSZ}$ = 0.06H or 0.17$H_{SF}$. Note that both dunes have the same bedform height but different steep face heights."

Lines 251 -259 have been rewritten as "In order to determine the extents of permanent and intermittent flow separations, the positions of the lines showing the 0% and 50% intermittency factor are calculated (Fig. 10).

For DUNE1_F1, an intermittent flow separation extending the entire steep lee side of the dune is detected. Below this intermittent flow separation, a permanent flow separation is limited only to the steep face of the dune.

Above DUNE2_F1, the intermittent flow separation becomes almost limited to the steep face and the permanent flow separation is significantly limited in extent compared to DUNE1_F1.

Above DUNE1 and when the flow is directed from the steep stoss to the gentle lee slope (i.e., DUNE1_F2), no permanent flow separation or intermittent flow separation are detected (Fig. 10c).

For DUNE2_F2, no permanent flow separation is detected over the gentle lee side of the dune. Interestingly, a small intermittent flow separation is detected over the short steep section (ca. x = 38 – 38.5 m) which has a slope 10°."

Lines 263-269 have now been rewritten as "Over DUNE1 and when the flow is directed from the gentle stoss to the steep lee slope (i.e., DUNE1_F1), a large and wide horizontal velocity gradient, $\partial \overline{u}/\partial z$, is observed to develop past the brink point and dissipate downstream of the steep lee side and over the gentle stoss side of the next dune (Fig. 11a). The thickest portion of the velocity gradient can be found at the steep face (ca. x = 36.0 m). This steep velocity gradient indicates the presence of a shear layer and significant vorticity.

For DUNE2_F1, a thin shear layer, almost attached to the bed and with a diffused-like structure, is observed (Fig. 11b).

Over DUNE1 and when the flow is now directed from the steep stoss to the gentle lee slope (i.e., DUNE1_F2), a very thin and weak shear layer can be seen to develop very close to the bed (Fig. 11c).

For DUNE2_F2, characteristics of an attached shear layer similar to that of DUNE1_F2 but with a slightly steeper velocity gradient within the short steep portion (10° slope) can be detected (ca. x = 38 – 38.5 m) (Fig. 11d)."

Lines 275 – 284 have been rewritten as "For DUNE1_F1, a well-defined turbulent wake can be seen developing just downstream of the brink point above the steep face (Fig. 12a). It propagates downstream of the steep side until it dissipates over the gentle stoss side of the next dune. The strongest portion of the wake with maximum TKE of 0.0089 $m^2/s^2$ is observed immediately downstream of the steep face (ca. x = 36.0 – 36.1 m).

For DUNE2_F1, the turbulence is further reduced with no defined wake structure and a more diffuse pattern can be detected (Fig. 12b). High TKE is concentrated within the immediate vicinity of the bed and diminishes towards the upper portion of the water column. Although not as strong as that of DUNE1_F1, high TKE occurs within the trough and the immediate downstream portion of the gentle stoss side.

When the flow is now directed from the steep stoss to the gentle lee slope of the dune (F2 flow), a similar trend of TKE is observed for both dunes, DUNE1_F2 and DUNE2_F2 (Figs. 12c & d). High TKE is concentrated in the near-bottom region of the flow with no appreciable wake structure. The TKE in the near-bottom flow region diminishes further when there is no steep face present at all (Fig. 11c).

Interestingly for DUNE2_F2 (Fig. 12d), a slightly elevated TKE can be detected within the short steep portion (ca. x = 38 – 38.5 m) of the gentle lee slope of the dune but the extent is still limited in the very near-bottom region of the flow."

Lines 289 – 305 have been rewritten as "For the four configurations tested in this study, the spatially-averaged Reynolds stresses, $<\tau_{uw}>$ ,generally increase towards the bed with rapid increase starting from the crest level and then decreases from the zero mean bed elevation down to the dune trough (Fig. 13).

The spatially-averaged Reynolds stresses are high when the flow is directed from the gentle stoss to the steep lee slope ( F1 flow, Figs. 13a & c). For DUNE1_F1, the maximum $<\tau_{uw}>$ is about 1.6 Pa at around z = -0.2 m. This means that the spatially-averaged turbulent stresses are strongly experienced just below mid-height of the dune.

For DUNE2_F1, the maximum $<\tau_{uw}>$ is reduced to about 0.6 Pa located near the trough (z = -0.4 m). This still imply that strong turbulent stresses are still occurring within the lower half of the dune height.

When the flow is now directed from the steep stoss to the gentle lee slope (F2 flow), the spatially-averaged Reynolds stress profiles for both dunes have an almost comparable vertical structure (Figs. 13b & d). Flow reversal (i.e., F2 flow) has reduced significantly the $<\tau_{uw}>$ magnitudes for both dunes. For DUNE1_F2, the maximum $<\tau_{uw}>$ is reduced to about 0.2 Pa almost at the zero mean bed elevation. This still implies that strong turbulent stresses are still occurring around the dune mid-height.

Similarly, for DUNE2_F2, the maximum $<\tau_{uw}>$ is also 0.2 Pa just above the zero mean bed elevation.

The spatially-averaged Reynolds stress profile can also provide a direct estimate of the total bottom shear stress as pointed out in previous studies (Nelson et al., 1993; Bennett and Best, 1995; Nikora et al., 2001; Mclean et al., 2008; Kwoll et al., 2016) by performing a regression analysis through the linear segment of the profile and projecting it downwards towards the zero mean bed elevation. The bed shear stress estimation shows a high coefficient of determination for all the linear fits (Fig. 13).

For DUNE1_F1, the linear fit is done above z = 0.2 m and from this linear fit, the estimated total bottom shear stress, $\tau_{o,}$ is 0.38 Pa.

For DUNE2_F1, the linear fit is above z = 0.34 m with estimated $\tau_o$ of 0.06 Pa.

When the flow is reversed and the flow is directed from the steep stoss to the gentle lee slope, a significant reduction in $\tau_o$ is observed. Both DUNE1_F2 and DUNE2_F2 yield a $\tau_o$ estimate of 0.04 Pa.

These findings show that flow bidirectionality can contribute to significant reduction of turbulence stresses regardless of the dune morphology."

Lines 311 – 326 have been rewritten as "The results of the quadrant analysis (Fig. 14) depict the percentages of observations for the four quadrant events. Quadrant 1 (Q1, outward interaction) events are fast water burst that moves upward and quadrant 3 (Q3, wallward, inward interaction) events are those slow water bursts that move downward. Both quadrants 1 and 3 are considered negative contributors to Reynolds stresses meaning that they contribute energy to the mean flow by extracting energy from turbulence such as shear layer vortices and coherent flow structures. Quadrant 2 (Q2, ejection) events are low-momentum near-bed fluid being thrown-up into the flow and quadrant 4 (Q4, sweep events) are those high-momentum fluid from above sweeping down into the flow. Both quadrants 2 and 4 are considered positive contributors to Reynolds stresses. Conversely, these turbulent events extract energy from the mean flow contributing to turbulence production.

For the four configurations tested in this study, percentages of observations for outward interaction (Q1) and wallward interaction (Q3) events show an increasing trend above the bed. Elevated percentages for these two events occur within z = 0.4-0.8 m above the bed for both dunes under both flow conditions (F1 and F2). On the other hand, percentages of observations for ejection (Q2) and sweep (Q4) events are highest near the bed and mid water column, especially for ejection events, and decrease toward the surface. Among the four quadrant events, the highest percentage of observations are observed mostly for ejection (Q2) events. High sweep (Q4) occurrences mainly take place in the very near-bottom flow region. These observations are common for both dunes under the two flow directions considered.

Some salient features specific to each dune and flow direction are also observed. For DUNE1_F1, a very intense and high occurrence of ejection (Q2) events are detected at the steep face of the dune and are being brought up further into the water column toward the water surface. These high Q2 occurrences ejected from the steep face are merging to a broader region of high Q2 occurrence located within z = 0.3-0.7 m from the bed. For DUNE2_F1, the same pattern seems to occur also although it is not as pronounced as the previous dune.

When the flow is directed from the steep stoss to the gentle lee slope (F2 flow), the spatial distribution of high ejection (Q2) occurrences changes for both dunes (i.e., DUNE1_F2 and DUNE2_F2). High Q2 occurrences are observed concentrating at the mid-water column around z = 0.2-0.4 m above the bed. Furthermore, high sweep (Q4) occurrences are also observed diminishing at the very near bottom for both dunes when the flow direction changes.”

3. In Section 4.1, the authors state: “The present findings demonstrate that both permanent flow separation and intermittent flow separation can exist for intermediate and low angle dunes depending on the lee side morphology, in particular the presence of a steep slope.” — I believe they are referring to F1 flows only, but it would be helpful if they could state that for clarity, since “stoss/lee slope” is flow-direction-dependent. For DUNE2_F2, based on the results/figures, there is no flow separation on the lee side (which would correspond to the gentle slope in the F2 case).

We have explicitly referred to both F1 and F2 cases when interpreting the results in Section 4.1. With this, Lines 370 – 403 have been rewritten as “Previous studies have pointed out the absence of permanent flow separation over low-angle dunes (Smith and Mclean, 1977; Kostaschuk and Villard, 1996; Roden, 1998; Carling et al., 2000; Best and Kostaschuk, 2002) and the possible presence of intermittent flow separation (Carling et al., 2000; Best and Kostaschuk, 2002). The present findings demonstrate that both permanent flow separation and intermittent flow separation can exist for both intermediate and low-angle dunes depending on the lee side morphology, in particular the presence of a steep slope. This is the case when the flow is directed from the gentle stoss to the steep lee slope of the dune (F1 flow).

While permanent flow separation is well documented over steep asymmetric high-angle dunes (Nelson et al., 1993; Bennett and Best, 1995; Roden, 1998; Kwoll et al., 2016), the permanent flow separation detected in this study especially for the intermediate-angle dune when the flow is directed from the gentle stoss to the steep lee

slope (DUNE1_F1) shows contrasting characteristics with typical large permanent flow separation (Best, 2005; Venditti, 2013; Lefebvre et al., 2014a, 2016). The observed permanent flow separation is more elongated and limited in extent. This small permanent flow separation only occupies the near-bottom flow region very close to the bed. Similar to previous observations above high-angle dunes (Bennett and Best, 1995; Kostaschuk, 2000), a small region at the steep face characterised by upward vertical velocity can also be detected. Because of the limited extent of the permanent flow separation above intermediate- and low-angle dunes when the flow is directed from the gentle stoss to the steep lee slope (DUNE1_F1 and DUNE2_F1), the flow separation lengths are much shorter compared to previously reported values typically between 4-6H for high-angle dunes (Engel, 1981; Paarlberg et al., 2007; Lefebvre et al., 2014; Naqshband et al., 2014) , 4.3H-6.5H$_{SF}$ for estuarine dunes (Carstensen and Holzwarth, 2023), 2.1-4.1H for 2D river dunes (Kwoll, 2013; Kwoll et al., 2016) and 5H$_{SF}$ for 3D river dunes (Lefebvre, 2019). The difference in the lengths of permanent flow separation can be attributed to the properties of the steep face (i.e., location and slope angle) as pointed out in previous studies (Lefebvre et al., 2016; Lefebvre, 2019; Lefebvre and Cisneros, 2023). The presence of a steep face is a controlling factor on the generation of flow separation. Over a steep slope such as the case when the flow is directed from the gentle stoss to the steep lee slope, a stronger adverse pressure gradient (i.e., $\partial p/\partial x \gg 0$) is encountered by the mean flow leading to a stronger and larger flow expansion which cause a permanent boundary layer separation. Such a process is also pointed out in a previous study about high and low-angle river dunes (Kwoll et al., 2016). On the contrary, only a weaker adverse pressure gradient is encountered over the gentle side of the dune which is not enough to form a permanent flow separation. This is especially true when the flow is directed from the steep stoss to the gentle lee slope (F2 flow).

   The intermittent flow separations that have been observed in this study have not been covered in much detail in previous studies. Specifically, we are able to show the presence and extent of intermittent flow separation for intermediate- and low-angle dunes. This study also confirms the previous claim that over low-angle dunes, an intermittent flow separation is present (Carling et al., 2000; Best and Kostaschuk, 2002) regardless of whether a permanent flow separation exists. Furthermore, our results demonstrate that even for low-angle dune possessing a very gentle mean slope but with some steeper portion, an intermittent flow separation can form. This is the case when the flow is directed from the steep stoss to the gentle lee slope of the dune (DUNE2_F2)."

Furthermore, we have also expanded the discussion of the flow separation to explain the influence of flow bidirectionality and its implication in the field settings. We have rewritten it as "This study has provided insights into the influence of flow bidirectionality on the flow and turbulence dynamics above dunes, particularly over intermediate- and low-angle tidal dunes. Flow bidirectionality effectively switches the flow dynamics between a more separated flow regime (F1 flow) and an attached near-bed flow regime (F2 flow) implying that flow separation metrics such as flow separation intermittency and size do not solely depend on morphology but also on flow orientation relative to the dune asymmetry. In the field, this can imply that flow

separation may depend strongly on the phase of the tidal cycle and the instantaneous flow separation metrics may not be generalized across the entire tide cycle. Furthermore, field implications highlight that bedforms and their associated flow and turbulence structures respond to changing forcing and can exhibit spatial and temporal variability, consistent with separation regimes that switch with flow reversal."

4. In Section 4.2 (lines 405–415), the authors summarize several criteria used in prior studies to define the extent of a turbulent wake. In the subsequent paragraph, they introduce the criterion adopted in this study, but it is unclear whether this approach is taken directly from Lefebvre et al. (2014) or newly proposed here. If the criterion is based on Lefebvre et al. (2014), it should be explicitly cited at the point where it is introduced. If it is the authors' own choice, it would be helpful to briefly justify why this particular threshold/definition was selected.

We now explicitly mentioned that our criterion was based from Lefebvre et al. (2014a). We have added this in Lines 418 – 420 as "In this study, we adopt the definition of Lefebvre et. al. (2014a) that the turbulent wake is the region enclosed by the isoline of 70% of the maximum TKE and once the wake has been delineated, the extent of the wake is defined as the horizontal distance between the farthest ends of this wake."

5. Overall, in the Discussion section, there seems to be a greater focus on ebb flows (i.e., when flow is going from the gentle to the steep side). There is relatively less discussion of what happens when "flow is opposed to dune asymmetry." Since bidirectionality is a central motivation, it would be valuable to expand the interpretation and implications of the F2 flows as well.

Thank you for this valuable suggestion. We have now added more interpretation as well as some implications of the F2 flow at each subsection of the Discussion section.

For Section 4.1, we have added a last paragraph for this point as "This study has provided insights into the influence of flow bidirectionality on the flow and turbulence dynamics above dunes, particularly over intermediate- and low-angle tidal dunes. Flow bidirectionality effectively switches the flow dynamics between a more separated flow regime (F1 flow) and an attached near-bed flow regime (F2 flow) implying that flow separation metrics such as flow separation intermittency and size do not solely depend on morphology but also on flow orientation relative to the dune asymmetry. In the field, this can imply that flow separation may depend strongly on the phase of the tidal cycle and the instantaneous flow separation metrics may not be generalised across the entire tide cycle. Furthermore, field implications highlight that bedforms and their associated flow and turbulence structures respond to changing forcing and can exhibit spatial and temporal variability, consistent with separation regimes that switch with flow reversal."

For Section 4.2, we have also added in the last paragraph a discussion on the influence of flow bidirectionality on turbulent wake. It was rewritten as "Similar with flow separation zone, flow bidirectionality again has considerable influence on the presence or absence of a turbulent wake. When the flow is directed from the steep stoss to the gentle lee slope, the turbulence structure transitions into a diffused, near-wall dominated TKE with no appreciable wake structure because the shear layer development is weak and largely attached to the bed. Our findings imply that in the field, turbulent wake formation is likely to be tide phase-dependent, with enhanced wake activity when the

flow is directed from the gentle stoss to the steep lee slope (represented by our F1 flow) and markedly reduced when the flow directed from the steep stoss to the gentle lee slope (represented by our F2 flow). Moreover, this phase dependency of wake structure can have further implication on the vertical mixing and suspension of sediments in natural flows (Kwoll et al., 2014)."

Similarly with Sections 4.1 and 4.3, we have also added in Section 4.3 a paragraph discussing the influence of flow bidirectionality on macroturbulent structures. We have added the paragraph as "Moreover, flow bidirectionality can also exert considerable influence on the organisation and strength of large-scale turbulence structures. Shear layer development and turbulence production that promote spatially coherent, large-scale turbulence structures are mostly pronounced over intermediate-angle dune and when the flow is directed from the gentle stoss to the steep lee slope. These features are, however, effectively damped when the flow is directed from the steep stoss to the gentle lee slope with the strongest suppression for the case of low-angle dune. This directionally-dependent modulation, similar to that happening for flow separation and turbulent wake, demonstrates that, depending on flow direction, a dune can switch between a macroturbulence-active regime (when the flow goes from the gentle to the steep side, as in our F1 flow conditions) and a largely attached, weakly coherent structure (when the flow goes from the steep to the gentle side, as in our F2 flow conditions), highlighting the importance of accounting for flow reversal in the flow dynamics over tidal dunes."

6.  The conceptual model should possibly be part of the Discussion, not the Results section, where it can read as though it is an additional empirical finding rather than a synthesis.

We have moved the conceptual model in the Discussion section and read it as if it is an additional empirical finding. We have largely rewritten it as "The conceptual diagram presented here demonstrates the significance of flow bidirectionality and dune morphology as controlling factors on the emergence of flow separation, detached shear layer and large-scale turbulence structure over low to intermediate-angle tidal dunes (Fig. 15). Our conceptual diagram emphasises that the same dune morphology can operate in two distinct dynamical regimes depending whether the flow is directed from the gentle stoss to the steep lee slope (as exemplified by our F1 flow) or directed from the steep stoss to the gentle lee slope (as exemplified by our F2 flow) which is a key feature of a bidirectional tidal flows.

Over the intermediate-angle tidal dune and when the flow is directed from the gentle stoss to the steep lee slope (DUNE1_F1), a clear partitioning of the flow structure and turbulence organisation is observed. A permanent flow separation is depicted as largely sweep-dominated (Q4 events) region near the bed while the overlying intermittent flow separation and turbulent wake region contain the more frequent and energetic ejections (Q2 events) that can rise upward and merge with other energetic structures advected from upstream dunes. This streamwise merging of highly energetic regions further support previous observations that influence of upstream bedforms can organise turbulence downstream, rather than individual dune behaving independently.

Over the low-angle dune still under F1 flow (DUNE2_F1), the same directional dependency of flow structures is maintained but the flow separation zones (permanent and intermittent flow separation) and energetic regions become damped. The permanent flow separation shrinks, the intermittent flow separation is narrow and the turbulence signature is weak with

the energetic ejection and sweep regions become restricted within the lower water column. With these findings from the two tested dune configurations, we demonstrate that dune morphology modulates the strength and extent of flow and turbulence structures.

The pronounced influence of flow directionality is evident when the flow is reversed and is directed from the steep stoss to the gentle lee slope of the dune (F2 flow). The directional dependency of the steep lee slope implies that the effective lee side becomes gentle (4°). In those cases, our observations show that there is any permanent flow separation but a small intermittent flow separation with a limited extent when a locally steep portion (10°) of the gentle slope is present (DUNE2_F2). Regions of steep velocity gradients and elevated turbulence are observed to be concentrated in the near-bottom region of the flow, consistent with a damped, attached shear layer. Correspondingly, the quadrant activity is weaker and less organised, with reduced presence of both ejection and sweep events relative to F1 flow.

Overall, our conceptual diagram highlights the influence of flow reversal compared to dune geometry that is especially relevant for large tidal dunes. When the flow is directed from the gentle stoss to the steep (low to intermediate-angle) lee slope, an active shear layer and a permanent flow separation develop that can sustain strong, vertically extensive large-scale turbulence structures. When the flow is reversed and is directed from the steep stoss to the gentle (low-angle) lee slope, there is no permanent flow separation, and the turbulent flow structure is shifted towards the bed with an attached shear layer and localised velocity gradients, suppressing the development of large-scale turbulent events."

7. The authors have done an excellent job in the Discussion section, especially in comparing the results to other studies. However, it would further strengthen the manuscript to explicitly discuss what the identified turbulence/separation structures imply for hydraulic roughness parameterizations in tidal settings, the implications for effective roughness arising from these bedforms, and real tidal flows that are not steady (e.g., how unsteadiness through a tidal cycle might modulate these patterns).

Thank you for these valuable suggestions. We have combined these questions with the additional questions also posed by Reviewer 1 because they have similarities. We devoted an additional sub-section in the Discussion section addressing these questions. The entire new subsection for this purpose was written as "4.5 Further implications on hydraulic roughness, superimposed dunes and tidal flows

Hydraulic roughness is an important parameter needed to quantify bed shear stress which in turn is needed for estimation of sediment transport. Our observations on flow separation and turbulence dynamics have implications on the effective hydraulic roughness arising from low to intermediate-angle dunes. The estimated total bottom shear stress shown in this study, which can also serve as a proxy for form roughness, is an order of magnitude larger over the intermediate-angle dune than the low-angle dune under the same flow condition (i.e., F1 flow). This sharp reduction in the total bottom shear stress for low-angle dune demonstrates how a lack of flow separation and strong turbulent wake translate to lower form drag and, thus, lowering the effective hydraulic roughness.

Our results can also have important implications for parameterisations of hydraulic roughness in tidal environments. Our findings show that even low-angle dunes (such as our DUNE2, with a gentle side of 4°), a flow separation and turbulence can still be generated suggesting

that previous hydraulic roughness estimators based solely on dune height or shape may not be adequate (De Lange et al., 2021). This is supported by previous field study which shows that dune size alone accounts only for 1/3 of the variance in hydraulic roughness in a river reach, indicating that other unresolved factors such as turbulence or flow divergence contribute significantly (De Lange et al., 2021). These findings point out the need for roughness parameterisations to account not only dune size but also detailed shape and flow reversals, especially for dunes found in natural tidal environments (Herrling et al., 2021).

Our tested dune configurations consist only of one scale of dune made of straight lines, and no superimposed secondary dunes were considered. Superimposed dunes, which are small bedforms that ride on the primary dunes, would likely modulate the flow separation and turbulence structures and alter the sediment transport dynamics above intermediate-angle and low-angle tidal dunes. Previous studies have shown how the steepness of the primary dune lee side controls the presence of secondary dunes over the lee side of the primary dune (Zomer et al., 2021). For our intermediate-angle dune when the flow is directed from the gentle stoss to the steep lee slope (DUNE1_F1), superimposed dunes cannot propagate further downstream owing to the steeper lee slope making the flow separation above the primary dune unaltered. For our low-angle dune under the same F1 flow (DUNE2_F1), the gentle lee slope (10°) allows the secondary dunes to propagate over the lee side which might break up the flow expansion zone and can suppress the already small flow separation forming above the primary dune (Dalrymple and Rhodes, 1995; Prokocki et al., 2022). From these two speculations on our tested dunes, it is clear that presence of secondary bedforms can effectively modify the primary dune effective lee side slope. Furthermore, these secondary dunes can also introduce their own micro-scale flow separation and turbulence which may collectively increase the total roughness (Zomer and Hoitink, 2024; Liu et al., 2025). Overall, the presence of superimposed dunes would induce additional form roughness and can either attenuate or enhance the primary dune flow separation and turbulence structures. This, in turn, would also influence turbulence and sediment flux over the primary dunes (Zomer and Hoitink, 2024).

Another key consideration in natural tidal flows is that they are unsteady, increasing and decreasing during each tidal phase. This unsteadiness may modulate our observed flow and turbulence structures within the tidal cycle. In natural tidal flows, the continually changing flow velocity and direction mean that flow separation and turbulence structure do not have much time to establish a fully developed steady state. Although our flow condition in the experiment is strictly steady unidirectional in two opposite directions which is an idealisation of the real tidal dynamics, we have effectively provided, at a particular time in the tidal cycle, an instantaneous snapshot of the flow and turbulence structures above our intermediate- and low-angle dunes which can serve as a guidance or reference for interpretation of the flow dynamics over natural tidal dunes under realistic tidal flows.

Finally, the present findings complement and help refine conclusions from previous field studies of dune and its associated hydraulic roughness. De Lange et al. (2021) showed that conventional dune geometry predictors underpredict the spatial variability of hydraulic roughness in rivers and that their attempt to correlate roughness with dune lee slopes was not satisfactory. Our experimental results suggest that even dunes with modest lee slopes produce flow separation and macroturbulent structures which might not be captured by simple dune geometry roughness predictors. These other factors could be the intermittent features

(intermittent flow separation and shear layer fluctuations) and flow phase dependency which might introduce roughness variability that is not apparent from dune morphology alone, explaining why lee slope metrics do not fully relate to roughness in field settings. Our controlled experiments, which use idealised representation of natural tidal dunes, validate previous conceptual framework on distinct regimes of dune morphology in a fluvial-tidal riverine setting (Prokocki et al., 2022), that in the tidal section of the river, dunes are mainly low-angle (ca. 10 - 15° lee slope) and generate only small flow separation with weaker wakes than high-angle dunes.

In summary, integrating our experimental results with field studies on tidal dunes (De Lange et al., 2021; Prokocki et al., 2022; De Lange et al., 2024) underscores that even intermediate to low angle dunes can still exert considerable impact to flow resistance. Their associated flow separation, if any, and turbulence characteristics must be accounted for to accurately predict hydraulic roughness and sedimental transport under realistic, unsteady tidal flow conditions."

**Citation**: https://doi.org/10.5194/egusphere-2025-4883-RC2